# EFFICIENTLY DISENTANGLING CLIP FOR MULTI-OBJECT PERCEPTION

## ABSTRACT

Vision-language models like CLIP excel at recognizing the single, prominent object in a scene. However, they struggle in complex scenes containing multiple objects. We identify a fundamental reason for this limitation: *VLM feature space exhibits excessive mutual feature information (MFI)*, where the features of one class contain substantial information about other, unrelated classes. This high MFI becomes evident during class-specific queries, as unrelated objects are activated alongside the queried class. To address this limitation, we propose DCLIP, an efficient framework that learns an optimal level of mutual information while adding only minimal learnable parameters to a frozen VLM. DCLIP uses two complementary losses: a novel MFI Loss that regulates class feature similarity to prevent excessive overlap while preserving necessary shared information, and the Asymmetric Loss (ASL) that aligns image features with the disentangled text features. Through this disentanglement, DCLIP reduces excessive inter-class similarity by 30%. On multi-label recognition, DCLIP performs favorably over SOTA approaches on VOC2007 and COCO-14 while using 75% fewer training parameters. For zero-shot semantic segmentation, it shows improved performance across six benchmark datasets. These results highlight the importance of feature disentanglement for multi-object perception in VLMs.

## 1 INTRODUCTION

Vision-language models (VLMs) such as CLIP have emerged as powerful tools for understanding visual content through natural language supervision. CLIP trains on a massive dataset of 400 million image-text pairs and demonstrates impressive performance in recognizing the salient object in the image, retrieving similar images from large datasets, and answering image-related natural language queries. However, an important question arises: *do these impressive capabilities transfer when CLIP processes images containing multiple objects*? As illustrated in Fig. 1, CLIP often struggles in such scenarios, failing to accurately recognize and localize all the objects present in the image. In this work, we investigate and identify the causes of this limitation and propose a framework that enables VLMs to handle complex images with multiple objects efficiently.

In our investigation, we analyze CLIP's features and identify two key factors contributing to this limitation. First, the spatial pooling operation in the visual encoder's final layer, while sufficient for identifying the prominent object, eliminates crucial spatial information needed to recognize and locate multiple distinct objects in an image. Second, and more importantly, we discover excessive entanglement between class features in the vision-language space, which we term mutual feature information (MFI). While some degree of feature similarity is useful for capturing broad semantic relationships (e.g., "dog" and "horse" share features as four legged animals), we find CLIP's features are excessively entangled. This entanglement (high MFI) becomes apparent during class-specific queries as illustrated in Fig. 1, where "dog" and "horse" regions also activate when we query "human." This activation pattern strongly correlates with the high similarity scores between class features (0.84 for human-horse, 0.80 for human-dog). We extend this analysis to the classes in VOC (Everingham et al., 2010) and COCO (Lin et al., 2014), where we observe average feature similarities of 0.77 in VOC and 0.69 in COCO (Tab. 3), confirming excessive feature entanglement in CLIP's feature space.

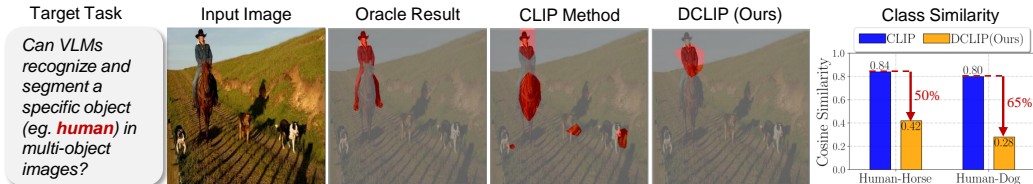

Figure 1: **Illustration of Feature Entanglement in CLIP.** Given the query "Human", CLIP exhibits high feature entanglement, spuriously activating regions corresponding to other objects (dogs and horse). In contrast, DCLIP (ours) accurately focuses only on the human. The rightmost plot quantifies this improvement through reduced cosine similarity between class features. **Takeaway:** By regulating inter-class similarity (Human-Horse: $0.84 \rightarrow 0.42$ [50% ↓], Human-Dog: $0.80 \rightarrow 0.28$ [65% ↓], DCLIP yields more disentangled features, crucial for precise multi-object perception.

To address the feature entanglement in VLMs, we propose DCLIP, a lightweight framework that disentangles class features using two complementary objectives. We draw inspiration from the redundancy reduction principle (Barlow et al., 1961), which states that sensory systems (eg. brain) recode input information such that redundancy is minimized without losing useful information, and extend it to the vision-language domain to reduce MFI. While previous approaches have focused on architectural modifications (Zhou et al., 2022a; Li et al., 2025; Bousselham et al., 2024) or prompt engineering (Sun et al., 2022; Rawlekar et al., 2025) to adapt VLMs for multi-object settings, they do not address the fundamental feature entanglement problem and often come with significant computational overhead. In contrast, DCLIP explicitly targets this root cause while adding only minimal learnable parameters on top of frozen VLMs, making it both effective and efficient. Our novel MFI Loss orthogonalizes text features to regulate inter-class similarity, preventing excessive overlap while preserving necessary shared information. Meanwhile the Asymmetric Loss (ASL) ensures proper cross-modal alignment. This joint training (MFI + ASL) produces an optimally disentangled feature space that significantly improves multi-object perception capabilities at low cost.

We evaluate DCLIP on multi-label recognition (MLR) and zero-shot semantic segmentation (ZS3). using established benchmarks: COCO-14 and VOC2007 (Everingham et al., 2010) for MLR. Importantly, for ZS3 evaluation, we use DCLIP projectors from MLR (trained on COCO-14) with image-level labels and evaluate it on six diverse datasets without any local annotations or fine-tuning. Our experimental results demonstrate that DCLIP reduces inter-class feature similarity by an average of 30% compared to CLIP across these datasets, leading to favorable MLR performance over SOTA methods on VOC2007 and the challenging COCO-14 dataset, while requiring 75% fewer parameters. For ZS3, DCLIP surpasses SOTA VLM methods, showing that reducing mutual feature information (MFI) is crucial for multi-object perception and can be achieved efficiently.

The main contributions of this work are:

- We identify that excessive mutual information between class features (MFI) is the bottleneck in VLMs' multi-object perception, leading to spurious cross-class activations

- We propose **DCLIP**, an efficient framework that regulates mutual information between classes through novel MFI (grounded in information bottleneck principle) and ASL losses, creating disentangled features while preserving image-text alignment

- We demonstrate DCLIP's feature improvements through two tasks: trained only for multi-label recognition with 75% fewer trainable parameters, it achieves strong MLR performance and outperforms prior work on six zero-shot semantic segmentation benchmarks

## 2 RELATED WORK

**Vision-Language Models for Multi-Object Perception.** Vision-language models (VLMs) trained with contrastive losses (Radford et al., 2021) are challenging to adapt for multi-object settings for two reasons: (1) Their reliance on global feature aggregation, which ignores local information. (2) The softmax operation in their training loss biases them toward single-object settings.

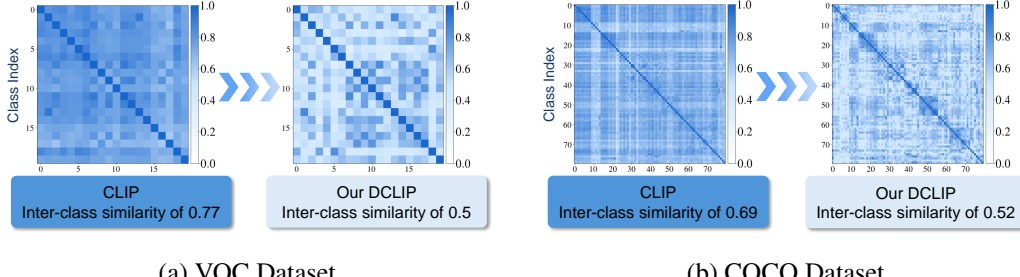

(a) VOC Dataset          (b) COCO Dataset

Figure 2: **Inter-Class Feature Similarity Analysis.** Comparison of class feature similarities between CLIP and DCLIP (our) across the VOC and COCO datasets. The heatmaps visualize the inter-class cosine similarity, where darker blue indicates higher similarity. **Takeaway:** DCLIP features demonstrate better separation as seen by reduced inter-class (off-diagonal entries) similarity values.

*Recognition.* Early efforts to adapt VLMs for recognition centered on learning prompts as classifiers for visual features (Zhou et al., 2022b). These methods were extended to multi-label settings by learning multiple prompts for each class (Sun et al., 2022; Hu et al., 2023). Subsequent works incorporated co-occurrence information to make predictions interdependent (Rawlekar et al., 2024). In contrast, our approach does not rely on prompt learning or co-occurrence modeling during pre-training. Furthermore, our features are adaptable to tasks beyond MLR.

*Localization.* Early approaches addressed localization by training image segmentation models and using VLMs to label the segmented regions (Kirillov et al., 2023). Later methods introduced pre-training setups that combined vision-language alignment with mask distillation to enhance localization (Dong et al., 2023). Recent works adapted features for localization without additional training by leveraging the spatial properties preserved in the value projection of CLIP's transformer-style aggregation (Zhou et al., 2022a). CLIP Surgery (Li et al., 2025) identified consistent noisy activations across classes and reduced them by subtracting average features from class-specific features (Li et al., 2025), though the cause of these activations remains unclear. In contrast, we provide a principled analysis identifying high mutual feature information (MFI) as the cause of poor multi-object perception, and propose a theoretically grounded solution through our MFI loss.

**Recoding information.** Shannon proposed that optimal information transmission involves designing codes with minimum entropy (Shannon, 1948). The redundancy reduction principle extended this idea to neuroscience, suggesting that sensory systems recode information to reduce redundancy with minimal loss (Barlow et al., 1961). This principle has since been applied to many recent works, including image compression (Ballé et al., 2016) and more popularly in representation learning (Oord et al., 2018; Chen et al., 2020; Zbontar et al., 2021; Henaff, 2020; He et al., 2020; Chen & He, 2021). While our loss function shares structural similarities with representation learning methods (a similarity and contrastive term), our method differs as follows: (1) DCLIP uniquely refines pre-trained VLM features, directly manipulating an existing, semantically rich space rather than learning representations from scratch (2) Crucially, our MFI loss operates on the fixed set of class text embeddings to directly reduce inter-class semantic similarity. This contrasts fundamentally with instance-discriminative contrastive losses that rely on intra-sample invariance (3) DCLIP is adapted for multi-object perception in VLMs, a setting prior works do not address.

## 3 DCLIP

### 3.1 ANALYSIS OF FEATURE ENTANGLEMENT IN CLIP

Before we introduce DCLIP, we first analyze the feature entanglement (high MFI) in CLIP that limits its multi-object perception capabilities. Our examination of pairwise cosine similarity between class text features reveals unexpectedly high inter-class similarities in CLIP's feature space, with average values of 0.77 for VOC and 0.69 for COCO classes (Fig. 2). Strikingly, even semantically distinct categories like "human" and "horse" show similarity scores as high as 0.84, far exceeding what their semantic relationship would suggest. This feature entanglement manifests visually when querying

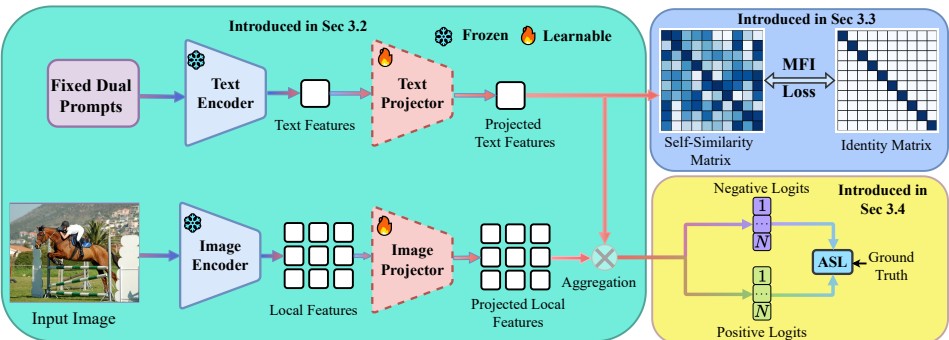

Figure 3: **DCLIP Overview.** Given image and class names in the dataset, CLIP extracts image and text features, which are projected by respective projectors to a disentangled space while preserving local image information. To reduce mutual feature information (MFI) between class features, we propose MFI loss that enforces the self-similarity matrix of projected text features to approximate an identity matrix, effectively regulating inter-class feature dependencies (Sec. 3.3). To align the image and the separated text features, we use the ASL loss in a multi-label recognition setup (Sec. 3.4). This setup aggregates the projected image and text features to obtain predicted logits (Sec. 3.4). The predicted logits are trained with ground truth labels using the widely used asymmetric loss (ASL) (Ridnik et al., 2021). Our training loss combines the ASL and MFI loss, the only trainable components are the projectors. During inference, we freeze CLIP's image and text encoders, along with our projectors for multi-label recognition and zero-shot semantic segmentation tasks.

for specific classes in complex scenes. As shown in Fig. 1, when querying for "human," CLIP erroneously activates "dog" and "horse" regions, a direct consequence of their entangled feature representations. The distribution of cosine similarity in Fig. 4 confirms this is not an isolated issue but rather a pervasive problem affecting the majority of class pairs. We attribute this entanglement to CLIP's contrastive training objective, which optimizes for global image-text alignment without explicitly constraining separation between different class features. Additionally, the global pooling operation mixes features across the entire image, which further exacerbates feature entanglement. Our analysis establishes a clear correlation between feature disentanglement and improved performance on multi-object perception (Fig. 7). As mutual feature information (MFI) decreases, both multi-label recognition and semantic segmentation performance consistently improve, confirming that regulating feature entanglement is crucial for enhancing CLIP's multi-object perception. This insight motivates our DCLIP framework, which maintains CLIP's rich semantic knowledge while explicitly reducing mutual information between class features to improve multi-object perception.

*DCLIP Overview.* Based on our analysis, we propose DCLIP, a framework that disentangles class features to enable effective multi-object perception. DCLIP leverages a pre-trained CLIP model ($f_\theta$), which comprises an image encoder ($f_{\theta,\text{img}}$) and a text encoder ($f_{\theta,\text{text}}$), both parameterized by $\theta$. All CLIP parameters ($\theta$) are frozen for all experiments. The DCLIP framework operates on multi-label dataset $\mathcal{D} = \{\mathbf{x}_i\,\mathbf{y}_i\}_{i=1}^{|\mathcal{D}|}$, where each image $\mathbf{x}_i$ is associated with a label vector $\mathbf{y}_i \in \{0,1\}^N$ indicating the presence of objects from multiple classes within our label space consisting of $N$ distinct classes $\{C_j\}_{j=1}^N$. DCLIP consists of three components: (1) Feature extraction and Projection, where we extract CLIP local features and project them into a disentangled space (Sec. 3.2), (2) Defining novel MFI Loss for disentangling text features (Sec. 3.3), and (3) Using ASL to align image features to the disentangled text features (Sec. 3.4).

## 3.2 Feature Extraction and Projection

We use CLIP as our feature extractor. Its image encoder ($f_{\theta,\text{img}}$) performs spatial pooling in the final layer, aggregating features from local regions into a $d$-dimensional vector for the input image $x_i$. However, as elaborated in Sec. 3.1, this pooling step, suppresses the contribution of less prominent objects, making it unsuitable for images with multiple objects. To mitigate this, we remove the final pooling layer of $f_{\theta,\text{img}}$ to preserve class-specific information across local regions. Then the encoder output for input ($\mathbf{x}_i$) is $f_{\theta,img}(\mathbf{x_i}) = \mathbf{z}_i \in \mathbb{R}^{H \times W \times d}$ , where $H$ and $W$ are the spatial dimensions.

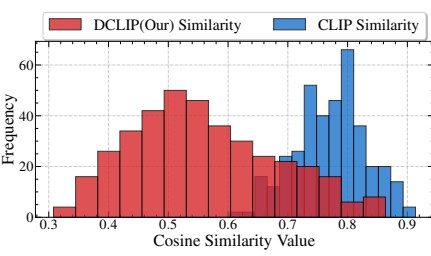 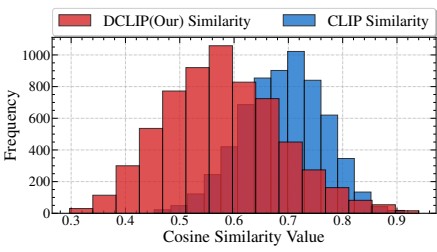

(a) VOC Dataset           (b) COCO Dataset

Figure 4: **Distribution of Inter-class Feature Similarities.** Histograms of cosine similarity values between class features for CLIP (blue) and DCLIP (red) across VOC (left) and COCO (right) datasets. **Takeaway:** DCLIP consistently shifts the distribution toward lower similarity values on both datasets, demonstrating significant feature separation between classes.

The text encoder remains unchanged. We use a fixed pair of positive and negative $(\mathbf{txt_{j,+}}, \mathbf{txt_{j,-}})$ prompts for each class $j$ as input to the text encoder. $\mathbf{txt_{j,+}}$ indicates the presence of the class in the image, while the $\mathbf{txt_{j,-}}$ indicates its absence. Passing these prompts through the text encoder produces $f_{\theta,text}(\mathbf{txt_i}) = \mathbf{t}_i$.

Following the feature extraction process, the resulting features (image ($\mathbf{z}_i$), text ($\mathbf{t}_i$)) lie in CLIP's feature space. As previously discussed in Sec. 3.1, this space exhibits significant feature entanglement and is not suitable for multi-object perception. To mitigate this, we project $\mathbf{z}_i$ and $\mathbf{t}_i$ into a new disentangled space using learnable projectors ($h_\phi : h_{\phi,\text{img}}$ and $h_{\phi,\text{text}}$), parameterized by weights $\phi$. These projectors map $\mathbf{z}_i$ and $\mathbf{t}_i$ from original space ($d$-dim) to a new disentangled space ($d'$-dim). Specifically, $h_{\phi,\text{img}}$ transforms $\mathbf{z}_i \rightarrow \mathbf{z}'_i$ ($\mathbb{R}^{H \times W \times d} \rightarrow \mathbb{R}^{H \times W \times d'}$) while preserving the spatial dimensions $(H, W)$. In addition, $h_{\phi,\text{text}}$ maps $\mathbf{t}_i \rightarrow \mathbf{t}'_i$ ($\mathbb{R}^d \rightarrow \mathbb{R}^{d'}$).

### 3.3 MFI LOSS

We design our projected feature space to regulate mutual feature information (MFI) between class features. To regulate MFI, we require isolated inputs for each class, as MFI measures the information shared between features of these isolated inputs. For images with multiple objects, obtaining isolated class inputs is challenging. For eg, an image containing both a horse and a dog would require computationally intensive segmentation methods to separate these objects. In contrast, text inputs are inherently isolated in our framework, as we already have individual class names (e.g., "horse" and "dog") available in the dataset. We leverage this natural isolation of text inputs by extracting text features, projecting them into our shared space to get $\mathbf{t}'$, and applying our proposed MFI loss to reduce information overlap between classes. Instead of using computationally expensive object segmentation for images, we propose an elegant solution that aligns these separated text features with image features using ASL loss (Sec. 3.4). **Theoretical grounding**: Our MFI loss is derived from the Information Bottleneck (IB) principle. We extend the standard IB formulation to explicitly account for inter-class information sharing:

$$\mathcal{IB} = I(Z_i; X_i) - \beta \left[ I(Z_i, Y_i) - I(Z_i, Y_{j \neq i}) \right] \tag{1}$$

where $I$ is the mutual information, $Z_i$, $X_i$, and $Y_i$ are the features, input, and target for class $i$. Under gaussian assumptions for high-dimensional entropy estimation (detailed proof in Appendix A), this reduces to minimizing cross-class correlations while preserving within-class structure. Our MFI loss:

$$\mathcal{L}_{\text{MFI}} = \underbrace{\sum_{i=1} (\mathbf{S}_{ii} - 1)^2}_{\text{Collapse Prevention}} + \lambda \underbrace{\sum_{i=1} \sum_{\substack{j=1 \\ j \neq i}} \mathbf{S}_{ij}^2}_{\text{MFI Reduction}} \tag{2}$$

where $\mathbf{S}$ is the self-similarity matrix and is given by $\mathbf{S}_{ij} = \mathbf{t}'^\top_i \mathbf{t}'_j$ where $\mathbf{t}'_i, \mathbf{t}'_j$ are the $i$-th and $j$-th column vectors of $\mathbf{t}'$ (i.e., $\mathbf{t}'_i, \mathbf{t}'_j \in \mathbb{R}^{1 \times d'}$). In this formulation, $\lambda$ is the hyperparameter that addresses the imbalance in the loss arising from the larger number of MFI reduction terms in $\mathbf{S}$

Table 1: **Comparison on multi-label recognition (MLR).** We compare the performance (Precision and mAP) and training efficiency (parameters, GPU hours) of our approach with SOTA VLM-based MLR methods on VOC2007 and COCO-14 datasets. DCLIP performs favorably over SOTA on VOC2007, and on the challenging COCO dataset, it outperforms SOTA while requiring only one-fourth of the parameters. red and blue indicate the best and the second best performance.

| Methods | VOC2007 | | | | COCO-14 | | | |
|---|---|---|---|---|---|---|---|---|
| | Param($\downarrow$) | GPU hrs($\downarrow$) | P($\uparrow$) | mAP($\uparrow$) | Param($\downarrow$) | GPU hrs($\downarrow$) | P($\uparrow$) | mAP($\uparrow$) |
| DualCoOp (Sun et al., 2022) | 0.3M | 3.6 | 80.5 | 94.2 | 1.3M | 16 | 72.9 | 83.6 |
| SCPNet (Ding et al., 2023) | - | >3.6 | – | 94.3 | 3.4M | 26 | – | 84.4 |
| TAI-DPT (Guo et al., 2023) | >0.3M | - | – | - | >1.3M | - | – | 84.5 |
| DualCoOp++ (Hu et al., 2023) | 0.4M | - | – | 94.9 | 1.5M | - | 76.3 | 85.1 |
| MLR-GCN (Rawlekar et al., 2024) | 0.4M | 3.6 | 81.1 | 94.4 | 1.3M | - | – | - |
| PositiveCoOp (Rawlekar et al., 2025) | 0.2M | 3 | 81.7 | 94.4 | 0.8M | 15 | 74.5 | 84.7 |
| DCLIP (Ours) | 0.4M | 3 | 85.6 | 95.4 | 0.4M | 13 | 80.1 | 85.6 |

compared to the collapse prevention terms. The MFI loss minimizes the inter-class similarity $\mathbf{S}_{ij}$ ($i \neq j$) while simultaneously preserving high intra-class $\mathbf{S}_{ii}$ to prevent feature collapse.

### 3.4 IMAGE-TEXT ALIGNMENT WITH ASL

**MLR Formulation.** MLR task involves identifying the subset of classes $\mathcal{C}_i \subseteq \{C_1, C_2, \ldots, C_N\}$ associated with the image $\mathbf{x}_i$. The goal is to learn a mapping function $g : \mathbf{x}_i \to \{-1, 1\}^N$, that maps input images to 1 if the class is present and $-1$ if the class is absent in the image.

We align $\mathbf{t}_i'$ and $\mathbf{z}_i'$ using the ASL loss in the MLR setup, eliminating the need for image segmentation. For each location $(h, w)$ in $(\mathbf{z}'_i)$, we detect the presence or absence of a class $j$, by computing the cosine similarity with positive text features ($\mathbf{t}'_{\mathbf{j},+}$) and negative text features ($\mathbf{t}'_{\mathbf{j},-}$):

$$\mathbf{l}_{ij}^+[h, w] = \mathbf{z}_i[h, w] \cdot \mathbf{t}'_{ij,+}, \quad \mathbf{l}_{ij}^-[h, w] = \mathbf{z}_i[h, w] \cdot \mathbf{t}'_{ij,-} \tag{3}$$

Higher similarity with positive text features indicates class presence, while higher similarity with negative features indicates absence. Following (Sun et al., 2022; Rawlekar et al., 2024; 2025), we apply softmax to focus on relevant regions ($q$) and aggregate to obtain final logits $p_i = [\mathbf{p}_i^+, \mathbf{p}_i^-]$:

$$\mathbf{q}_i^{\pm}[h, w] = \frac{\exp(\mathbf{l}_i^{\pm}[h, w])}{\sum_{h', w'} \exp(\mathbf{l}_i^{\pm}[h', w'])}, \quad \mathbf{p}_i^{\pm} = \sum_{h, w} \mathbf{q}_i^{\pm}[h, w] \cdot \mathbf{l}_i^{\pm}[h, w] \tag{4}$$

See Appendix B for details. We train using Asymmetric Loss (ASL) (Ridnik et al., 2021). Here, $p_i^j$ is the prediction for label $y_i^j$, and $p_{i,\delta}^j = \max(\hat{y} - \delta, 0)$ with shifting parameter $\delta$.

$$\mathcal{L}_{ASL}(p_i^j) = \begin{cases} \left(1 - p_i^j\right)^{\gamma_+} \log\left(p_i^j\right), & \text{if } y_i^j = 1, \\ \left(p_{i,\delta}^j\right)^{\gamma_-} \log\left(1 - p_{i,\delta}^j\right), & \text{otherwise} \end{cases} \tag{5}$$

**Training.** Our training objective is composed of two components: (1) mutual feature information loss that enforces the separation between class text features and (2) Asymmetric loss function (Ridnik et al., 2021), designed for MLR that aligns the image features and text features to obtain predictions for an image. Here, $\alpha$ controls the relative importance of the two objectives

$$\mathcal{L}_{\text{DCLIP}} = \mathcal{L}_{\text{ASL}} + \alpha \mathcal{L}_{\text{MFI}} \tag{6}$$

## 4 EXPERIMENTS

In this section, we describe the datasets, evaluation metrics, implementation details, and performance analysis for multi-label recognition (MLR) and zero-shot semantic segmentation (ZS3).

### 4.1 DATASETS AND METRICS

1) Adaptation with MLR: We evaluate the MLR performance using mean-Average Precision (mAP) on datasets: (1) **COCO-14** (80 classes) (Lin et al., 2014). Following recent works (Sun et al., 2022;

Table 2: **Zero-shot semantic segmentation (ZS3) comparison.** We compare DCLIP with other SOTA methods using mIoU metric. The abbreviations are: Loc Ann.+ FT: local annotations and fine-tuning, Bkgd: include background class, No Bkgd: ignore background, red and blue indicate the best and the second best performance.

| Method | Loc Ann. + FT | VOC12 Bkgd | COCO-17 Bkgd | COCO-17 No Bkgd |
|---|---|---|---|---|
| SPNet (Xian et al., 2019) | ✓ | 15.6 | - | - |
| ZS3Net (Bucher et al., 2019) | ✓ | 17.7 | - | - |
| CLIP-ES (Lin et al., 2023) | ✓ | 75.0 | - | - |
| CLIP (Radford et al., 2021) | ✗ | 14.1 | 3.9 | 5.6 |
| CLIPSurgery (Li et al., 2025) | ✗ | 17.5 | 13.0 | 22.9 |
| CLIP-VV (Li et al., 2025) | ✗ | 32.6 | 19.9 | 35.5 |
| | ✗ | 36.0 | 22.7 | 37.8 |

Table 3: **MFI Reduction.** MFI values for VOC and COCO. DCLIP significantly reduces MFI.

| Method | VOC | COCO |
|---|---|---|
| CLIP | 0.77 | 0.69 |
| DCLIP | 0.50 | 0.52 |
| $\Delta$ (%) | 34.8 | 24.9 |

Table 4: **MFI Loss Ablation.** Without MFI, MLR performance drops by 1.2 mAP on COCO.

| Method | ASL | MFI | mAP |
|---|---|---|---|
| Ours | ✓ | ✗ | 84.2 |
| | ✓ | ✓ | 85.4 |

Rawlekar et al., 2024; 2025), we train on the training set and evaluate on the validation set. (2) **VOC2007** (20 classes) (Everingham et al., 2010). Following (Sun et al., 2022; Rawlekar et al., 2024; 2025), we use the train-val set for training and the test set for evaluation.

2) Zero-Shot Semantic Segmentation (ZS3): We extract projectors trained for MLR on COCO-14 (image-level labels) and evaluate ZS3 using the mIoU metric on the following datasets: PASCAL VOC 2012 (20 classes + background) (Everingham et al., 2010), COCO-2017(80 classes + background) (Lin et al., 2014), cityscapes (30 classes) (Cordts et al., 2016), context (59 classes) (Mottaghi et al., 2014), stuff (91 classes) (Caesar et al., 2018) and ADE20k (150 classes) (Zhou et al., 2017)

## 4.2 IMPLEMENTATION DETAILS

We use CLIP's (Radford et al., 2021) original pre-trained encoder weights for all our experiments and keep them frozen. Consistent with popular MLR and ZS3 literature, we use a ResNet-based visual encoder and the standard transformer for text encoding (Sun et al., 2022; Ding et al., 2023; Hu et al., 2023; Rawlekar et al., 2025; 2024; Guo et al., 2023; Li et al., 2025; Lin et al., 2023). We conduct all experiments on a single RTX A4000 GPU. For **MLR**(Sec. 3.4) we follow the settings and hyperparameters from recent works (Sun et al., 2022; Rawlekar et al., 2024; 2025). This includes resizing images to $448$, applying Cutout (DeVries & Taylor, 2017) and RandAugment (Cubuk et al., 2020) transforms. Our projectors ($h_\phi$) are implemented as multi-layer perceptrons. Specifically, the image projector follows a [$512 \rightarrow 256$] architecture, while the text projector is designed as [$512 \rightarrow 384 \rightarrow 256$] with batch normalization and ReLU. We train both projectors with stochastic gradient descent with an initial learning rate of 0.002, which is reduced by cosine annealing. We train the DCLIP for 50 epochs with a batch size of 32. We follow (Sun et al., 2022; Rawlekar et al., 2024; 2025), and use ASL hyperparameters in eq. 5 as $\gamma_- = 2$, $\gamma_+ = 1$ and $\delta = 0.05$. We set $\lambda = 0.2$ and $\alpha = 7e-5$ when pre-trained with COCO-14 in eq. 2. For **Zero-Shot Semantic Segmentation**, we adopt the vv attention (Li et al., 2025) that prevents inversion of activation commonly observed in CLIP. We then add our pre-trained projectors to CLIP. To obtain the segmentation mask, we compute the cosine similarity between locally projected image features ($\mathbf{z}'$) and projected text features for all classes in the dataset. We use the template "A photo of a {classname}." Lastly, we use bilinear interpolation to upsample the mask to the input image size.

## 4.3 RESULTS

**Multi-Label Recognition.** We compare DCLIP with other SOTA VLM-based MLR approaches. In Tab. 1, we present a detailed comparison of the performance (mAP-averaged over five runs), number of training parameters, and GPU hours required by each method on the VOC2007 (Everingham et al., 2010) and COCO-14 (Lin et al., 2014) datasets. For VOC 2007, we observe that DCLIP performs favorably over DualCoOp++(Hu et al., 2023), requiring equal parameters but fewer training hours. Additionally, on the more challenging COCO-14 dataset, DCLIP outperforms DualCoOp++ while requiring 75% fewer training parameters and fewer training hours on an NVIDIA A4000 GPU.

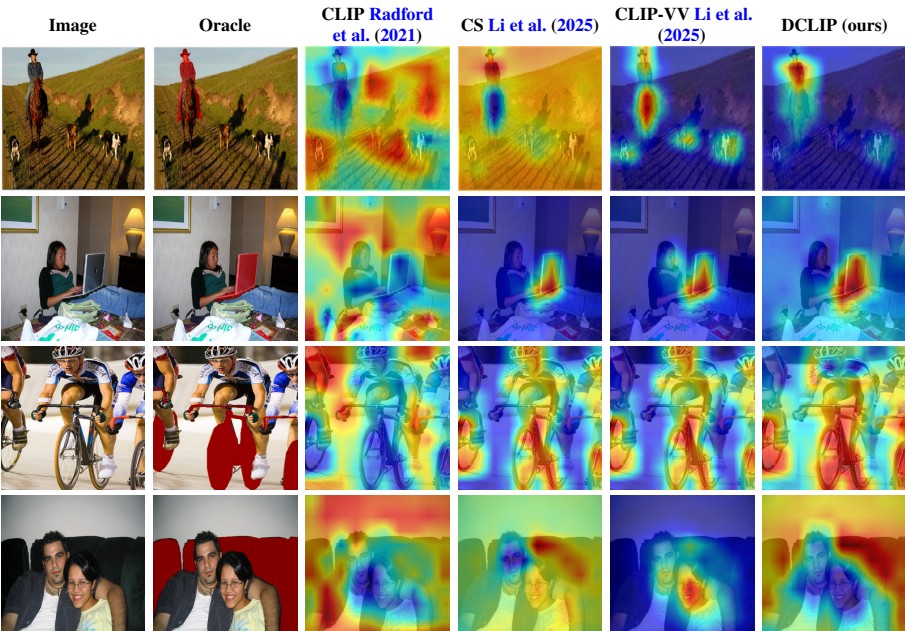

| Image | Oracle | CLIP Radford et al. (2021) | CS Li et al. (2025) | CLIP-VV Li et al. (2025) | DCLIP (ours) |

Figure 5: **Qualitative ZS3 Comparison.** Visualization of ZS3 results for CLIP (Radford et al., 2021), CLIP Surgery (CS) (Li et al., 2025), CLIP-VV (Li et al., 2025), and DCLIP (ours) across multiple categories. The red regions in Oracle show the queried classes. The heatmaps for different methods show activation regions for each queried class, where darker red indicates strongly activated regions. DCLIP (ours) produces more separated activations, leading to improved class localization.

**Zero-Shot Semantic Segmentation.** We categorize our comparisons into two groups. The first group includes approaches that use local annotations (segmentation masks) to fine-tune the network (Xian et al., 2019; Bucher et al., 2019; Lin et al., 2023). The second group does not use any local annotations (Radford et al., 2021; Li et al., 2025). As **we do not use any form of local annotations**, DCLIP belongs to the second group. Our projectors train only on image-level MLR labels. Results are summarized in Tab. 2. For VOC2012, we report the mIoU with background. Following (Bousselham et al., 2024), we use a threshold of 0.85 to identify the background. Our approach performs favorably over CLIP Surgery by 18.5 mIoU and CLIP-VV by 3.4 mIoU on VOC2012. On COCO-17, our method outperforms CLIP Surgery and CLIP-VV by 9.7 and 2.8 mIoU with background, and by 14.9 and 2.3 mIoU without background. Fig. 6 shows performance on cityscapes (Cordts et al., 2016), context (Mottaghi et al., 2014), stuff (Caesar et al., 2018) and ADE20k (Zhou et al., 2017). This demonstrates DCLIP's ability to learn domain-agnostic disentangled features that transfer across diverse visual environments (urban scenes, indoor scenes, natural images).

## 5 ANALYSIS

**Feature Disentanglement and its Impact.** This section examines both quantitative and qualitative impacts of class feature disentanglement. In Fig. 2, we visualize self-similarity matrices for class text features, revealing that DCLIP achieves substantially lower inter-class similarity (off-diagonal values) than baseline CLIP. The distribution of similarity values in Fig. 4 further illustrates how DCLIP shifts feature representations toward reduced inter-class similarity. We quantitatively assess this disentanglement in Tab. 3 using average inter-class similarity, where DCLIP consistently demonstrates lower values across datasets, confirming effective feature separation. Crucially, Fig. 7 establishes a clear inverse relationship between MFI and performance: as MFI decreases, indicating greater feature disentanglement, we observe gains in both multi-label recognition and zero-shot semantic segmentation. Our ablation experiments provide additional validation, showing that removing the MFI Loss component for separation results in a significant 1.2 mAP reduction in MLR task performance (Tab. 4). These results strongly confirm that disentangling class features is both necessary and beneficial for improving performance for multi-object perception.

Table 5: **CLIP's Multi-Label Recognition with DCLIP Segments.** By segmenting images into individual objects using DCLIP and processing each segment through CLIP, we achieve consistent improvements on VOC2007 and COCO-14.

| Dataset | Backbone | CLIP (mAP) | + DCLIP Segments (mAP) |
|---------|----------|------------|------------------------|
| VOC2007 | RN101 | 78.73 | **80.71** |
|         | RN50  | 76.20 | **79.87** |
| COCO-14 | RN101 | 50.10 | **52.00** |
|         | RN50  | 47.30 | **50.15** |

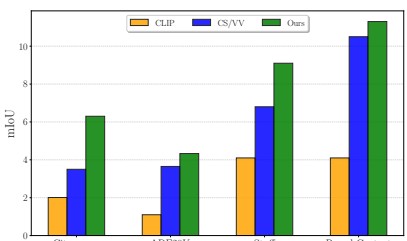

Figure 6: ZS3 results for Cityscapes, Pascal Context, Stuff and ADE20K dataset.

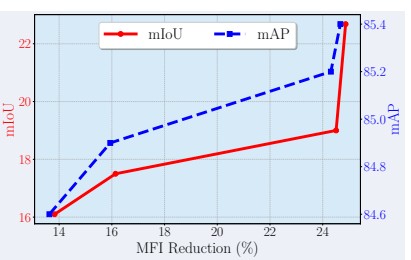

Figure 7: **Performance vs. MFI Reduction.** Increase in class feature separation (i.e., MFI decreases) improves performance on MLR (mAP) and ZS3 (mIoU) tasks.

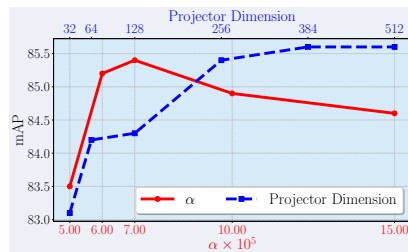

Figure 8: **Hyperparameter Ablations.** 1) DCLIP demonstrates stability across $\alpha$ variations. 2) Increasing projector dimension improves performance (saturates beyond 256 dims).

**DCLIP's Segments for Multi-Label Recognition.** To evaluate whether DCLIP produces meaningful object segments, we test its utility in improving CLIP's MLR performance. We reformulate the MLR problem by combining global and local predictions. For global predictions, we pass the input image directly through the original CLIP model. However, as discussed in Sec. 1, these predictions are often dominated by more prominent objects, ignoring smaller objects in multi-object scenes. To complement this, we generate local predictions by first segmenting the image using DCLIP to isolate individual objects, then processing each segment independently through the original CLIP model. The predictions from all segments and global image are combined to get final scores. The results in Tab. 5 on VOC2007 and COCO-14 demonstrate that DCLIP extracts meaningful object segments.

**Ablation.** We study the sensitivity of MLR performance to the sensitivity of hyperparameters and projector choices. (1) $\alpha$ (ASL–MFI Trade-off): Fig. 8 shows that varying $\alpha$, the coefficient that balances the ASL and MFI loss, has minimal impact on performance, indicating low sensitivity to $\alpha$. (2) $\lambda$ (MFI Loss Weighting): We varied $\lambda$, which controls the trade-off between collapse prevention and redundancy reduction in the MFI loss (Eq. 2), across the range $[0.02, 0.2]$. Performance remained stable throughout, suggesting insensitivity to $\lambda$ (Appendix Sec D.1). (3) Projector Dim: Increasing the dimension of the projector beyond 256 dim led to saturation (Fig. 8). Ablations on loss (BCE vs Focal vs ASL), pooling–projection order and architectures are in Appendix Sec. D.2, D.3, D.4.

## 6 CONCLUSIONS

In **conclusion**, our work identifies that high mutual feature information (MFI) between class features impairs CLIP's ability for multi-object perception. To address this, we propose DCLIP, an efficient framework that regulates CLIP features entanglement using our proposed MFI loss and the ASL loss. Experiments across multiple benchmarks show that reducing feature entanglement significantly improves multi-label recognition and zero-shot segmentation performance. These results establish feature disentanglement as essential for adapting VLMs to scenes with multiple objects. **Limitations** of our approach include handling fine-grained subcategories within the same superclass (e.g., different dog breeds or species of birds). This limitation partially stems from CLIP's inherent limitation to fine-grained discrimination. Future work could explore adaptive disentanglement strategies that operate at multiple semantic levels to support hierarchical concepts.

## 7 REPRODUCIBILITY STATEMENT

We have taken several steps to ensure reproducibility of our work. All dataset splits, hyperparameters, and training settings (including optimizer type and learning rate schedules) are described in detail in the Implementation Details section. These details are sufficient to reproduce our results independently of the released code. In addition, we provide algorithmic pseudo-code in Appendix C for clarity, which outlines the entire training pipeline step-by-step.

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

# Technical Appendix: Disentangling CLIP for Multi-Object Perception

## A  MUTUAL FEATURE INFORMATION LOSS AND INFORMATION BOTTLENECK PRINCIPLE

In this section, we relate our Mutual Feature Information (MFI) loss to the Information Bottleneck (IB) principle Tishby & Zaslavsky (2015).

### A.1  MUTUAL FEATURE INFORMATION (MFI) LOSS RECAP:

$$\mathcal{L}_{\text{MFI}} = \underbrace{\sum_{i=1} \left(\mathbf{S}_{ii} - 1\right)^2}_{\text{Collapse Prevention}} + \lambda \underbrace{\sum_{i=1} \sum_{\substack{j=1 \\ j \neq i}} \mathbf{S}_{ij}^2}_{\text{MFI Reduction}} \tag{7}$$

where $\mathbf{S}$ is the self-similarity matrix obtained from $\mathbf{t}'$. Here, $\mathbf{S}$ is defined by

$$\mathbf{S}_{ij} = {\mathbf{t}'}_i^\top \mathbf{t}'_j, \quad \forall i, j$$

where $\mathbf{t}'_i, \mathbf{t}'_j$ are the $i$-th and $j$-th column vectors of $\mathbf{t}'$ (i.e., $\mathbf{t}'_i, \mathbf{t}'_j \in \mathbb{R}^{d'}$). In this formulation, $\lambda$ is the hyperparameter that addresses the imbalance in the loss arising from the larger number of MFI reduction terms in $\mathbf{S}$ compared to the collapse prevention terms.

### A.2  INFORMATION BOTTLENECK (IB) PRINCIPLE:

The IB principle was introduced to extract relevant information from an input random variable $\mathbf{X}$ about an output random variable $\mathbf{Y}$. This relevant information is defined as mutual information $I(X; Y)$. The relevant part of $\mathbf{X}$, is given by $\mathbf{Z}$. The principle assumes a chain $\mathbf{X} \to \mathbf{Z} \to \mathbf{Y}$ with the goal to minimize mutual information $I(Z; X)$ and maximize $I(Z; Y)$

$$\mathcal{IB} = I\left(Z; X\right) - \beta I\left(Z; Y\right) \tag{8}$$

Here $\beta$ captures the tradeoff between the two terms. For neural networks, $\mathbf{X}$ represents the input, $\mathbf{Z}$ are its features, and $\mathbf{Y}$ is the output.

### A.3  FORMULATION:

We observe that CLIP's feature space suffers from feature entanglement, where representations of one object(class) inadvertently contain information about other objects. To address this, we seek to enforce that CLIP features for each class contain only relevant information for that specific class while suppressing information about all other classes.

We achieve this, we expand the information bottleneck formulation in eq. 8 by explicitly accounting for inter-class information sharing:

$$\mathcal{IB} = I(Z_i; X_i) - \beta \left[ I(Z_i, Y_i) - I(Z_i, Y_{j \neq i}) \right] \tag{9}$$

where $X_i$, $Z_i$, and $Y_i$ represent the input, learned features, and target output for class $i$, respectively.

Our modified information bottleneck ($\mathcal{IB}$) objective achieves disentangled features through three components:

1. Minimizing $I(Z_i; X_i)$ enforces the features $Z_i$ to retain only the essential information from input $X_i$ necessary for predicting $Y_i$, discarding irrelevant details

2. Maximizing $I(Z_i; Y_i)$ ensures that features $Z_i$ are highly informative for predicting the correct class $Y_i$, promoting discriminative representations.

3. Minimizing $I(Z_i; Y_j)$ for $j \neq i$, reduces mutual information between class features, preventing features of class $i$ from encoding information about other classes $j$, thus disentangling features.

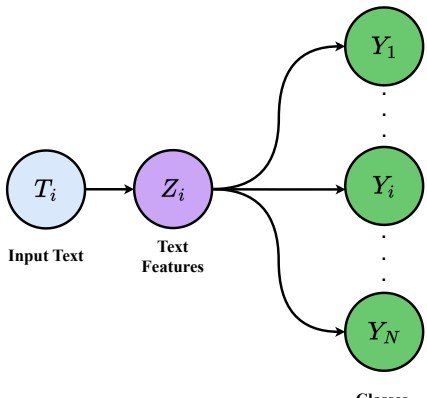

Figure 9: The Information Bottleneck principle is applied for feature disentanglement. Given an input text $T_i$, the encoder produces text features $Z_i$, which contain information about the output classes $Y_i$. Our objective is to ensure that $Z_i$ retains only the information necessary to map to its corresponding class $Y_i$ while minimizing its information about other classes $Y_j$ ($j \neq i$)

## A.4   DERIVATION:

To minimize the objective in Eq. 9, we express mutual information ($I$) in terms of entropy ($H$) using the standard identity:

$$I(A; B) = H(A) - H(A|B) \tag{10}$$

where $H(A)$ is the marginal entropy of $A$ and $H(A|B)$ is the conditional entropy of $A$ given $B$.

Substituting this into our objective yields:

$$\mathcal{IB} = [H(Z_i) - H(Z_i|X_i)] - \beta\left[\cancel{H(Z_i)} - H(Z_i|Y_i) - \cancel{H(Z_i)} + H(Z_i|Y_j)\right] \tag{11}$$

Following the established approach in neural information bottleneck theory Zbontar et al. (2021), during objective evaluation we treat our projection function $h_\phi$ as deterministic , making the conditional entropy $H(Z_i|X_i) = 0$. This standard assumption is justified in our case because our projectors consist of linear layers and ReLU activations without any stochastic components (no dropout or batch normalization during inference), ensuring that for any given $X_i$, the projected representation $Z_i$ is perfectly determined. This deterministic treatment allows tractable analysis while the key insight: reducing mutual information between class features—remains valid regardless of the specific entropy computation details. This simplifies our objective to:

$$\mathcal{IB} = H(Z_i) + \beta\left[H(Z_i|Y_i) - H(Z_i|Y_j)\right] \tag{12}$$

Estimating entropy in high-dimensional spaces is computationally intractable, requiring exponentially many samples as dimensionality increases Hall & Morton (1993); Greenewald et al. (2023). Following established approaches Robin & Scrucca (2023); Greenewald et al. (2023); Goldfeld et al. (2018), we make the assumption that features $\mathbf{Z}$ follow a Gaussian distribution. This assumption is standard in information-theoretic analysis of deep representations and is particularly justified for our normalized projected features.

For a Gaussian distribution $\mathbf{x} \sim \mathcal{N}(\boldsymbol{\mu}, \boldsymbol{\Sigma})$ with $\boldsymbol{\mu} \in \mathbb{R}^d$ and $\boldsymbol{\Sigma} \in \mathbb{R}^{d \times d}$, the entropy is:

$$\mathcal{H}(\mathbf{x}) = \frac{1}{2} \log\left[(2\pi e)^d |\boldsymbol{\Sigma}|\right] = \frac{d}{2} + \frac{d\log(2\pi)}{2} + \frac{1}{2}\log|\boldsymbol{\Sigma}| = C + \frac{1}{2}\log|\boldsymbol{\Sigma}|$$

Substituting into Eq. 12 and ignoring constant terms that don't affect optimization:

$$\mathcal{IB} \propto \log|\boldsymbol{\Sigma}_{Z_i}| + \beta\left[\log|\boldsymbol{\Sigma}_{Z_i|Y_i}| - \log|\boldsymbol{\Sigma}_{Z_i|Y_j}|\right] \tag{13}$$

Similar to Zbontar et al. (2021), to make the optimization tractable, we work directly with the matrix elements rather than their determinants.

$$\mathcal{IB} \propto \mathbf{\Sigma}_{Z_i} + \beta \left[ \mathbf{\Sigma}_{Z_i|Y_i} - \mathbf{\Sigma}_{Z_i|Y_j} \right] \tag{14}$$

For simplicity of understanding, we decompose the covariance matrix into diagonal and off-diagonal components:

$$\mathbf{\Sigma}_{Z_i} = \text{diag}(\mathbf{\Sigma}_{Z_i}) + \text{off-diag}(\mathbf{\Sigma}_{Z_i})$$

For notational convenience in the following derivation, we denote $\text{diag}(\mathbf{\Sigma}_{Z_i})$ by $\mathbf{\Sigma}_{Z_i|Y_i}$ and off-diag$(\mathbf{\Sigma}_{Z_i})$ by $\mathbf{\Sigma}_{Z_i|Y_j}$, where $\mathbf{\Sigma}_{Z_i|Y_i}$ captures individual feature variances (diagonal entries) and $\mathbf{\Sigma}_{Z_i|Y_j}$ captures cross-feature correlations (off-diagonal entries). Substituting this decomposition:

$$\mathcal{IB} \propto \left[ \mathbf{\Sigma}_{Z_i|Y_i} + \mathbf{\Sigma}_{Z_i|Y_j} \right] + \beta \left[ \mathbf{\Sigma}_{Z_i|Y_i} - \mathbf{\Sigma}_{Z_i|Y_j} \right] \quad = (1+\beta)\mathbf{\Sigma}_{Z_i|Y_i} + (1-\beta)\mathbf{\Sigma}_{Z_i|Y_j} \tag{15}$$

As the features $Z_i$ are normalized (Algorithm 2-L16), $\mathbf{\Sigma}_{Z_i|Y_i} = \mathbf{I}$ (identity matrix), the diagonal entries are 1 and this term becomes constant and can be ignored during optimization. This yields our final objective:

$$\mathcal{IB} \propto \mathbf{\Sigma}_{Z_i|Y_j} \tag{16}$$

Our optimization problem becomes:

$$\begin{aligned} \min \quad & \mathbf{\Sigma}_{Z_i|Y_j} \\ \text{subject to} \quad & \mathbf{\Sigma}_{Z_i|Y_i} = \mathbf{I} \quad \forall i \end{aligned} \tag{17}$$

To implement this theoretical optimization in practice, we translate it to our inner product matrix formulation. Our optimization becomes minimizing off-diagonal entries $S_{ij}$ while maintaining diagonal entries $S_{ii} = 1$.

Since our approach explicitly normalizes all projected text features $\|\mathbf{t}'_i\| = 1$, the constraint $S_{ii} = 1$ is automatically satisfied for our inner product matrix $\mathbf{S}$ where $S_{ij} = \mathbf{t}'^{\top}_i \mathbf{t}'_j$.

$$\mathcal{L}_{MFI} = \sum_i (S_{ii} - 1)^2 + \lambda \sum_{i \neq j} S_{ij}^2 \tag{18}$$

where the first term maintains $S_{ii} = 1$ (enforcing normalization) and the second term minimizes cross-class inner products $S_{ij}$ (implementing the theoretical objective of feature orthogonalization).

### A.5   MFI IMPLEMENTATION:

1. Our encoder architecture combines CLIP's pre-trained encoder with a learnable projector (detailed in Section 3.2). This design allows us to preserve CLIP's rich semantic knowledge by keeping the original encoder frozen, while the projector learns to disentangle features through MFI loss optimization.

2. Empirically, we found that computing feature correlations along the larger dimension yields superior results. Specifically, we use $\mathbf{S}_{ij} = \mathbf{t}'^{\top}_i \mathbf{t}'_j$ for the cross-correlation matrix, where this formulation captures inter-feature relationships more effectively.

## B   IMAGE-TEXT ALIGNMENT WITH MLR

In this section, we elaborate on the local image-text alignment mechanism based on ASL, as a continuation of Sec. 3.4.

The projected image features for the image $x_i$ is $\mathbf{z}'_i \in \mathbb{R}^{H \times W \times d'}$, and disentangled text features $\mathbf{t}'_i \in \mathbb{R}^{2N \times d'}$ be the corresponding disentangled text features, where $N$ is the number of classes.

The text features is combined using positive and negative prompts:

$$\mathbf{t}'_i = [\mathbf{t}'_{i,+}, \mathbf{t}'_{i,-}], \quad \mathbf{t}'_{i,+} = [\mathbf{t}'_{ij,+}]_{j=1}^N, \quad \mathbf{t}'_{i,-} = [\mathbf{t}'_{ij,-}]_{j=1}^N.$$

For each class $j$, we compute the local positive and negative logits at every spatial location $(h, w)$:

$$\mathbf{l}_{ij}^+[h, w] = \mathbf{z}_i[h, w] \cdot \mathbf{t}'_{ij,+}, \quad \mathbf{l}_{ij}^-[h, w] = \mathbf{z}_i[h, w] \cdot \mathbf{t}'_{ij,-}.$$

We collect the per-class logits into tensors:

$$\mathbf{l}_i^+ = [\mathbf{l}_{ij}^+]_{j=1}^N, \quad \mathbf{l}_i^- = [\mathbf{l}_{ij}^-]_{j=1}^N.$$

Next, we compute softmax maps for the logits logits:

$$\mathbf{q}_i^+[h, w] = \frac{\exp(\mathbf{l}_i^+[h, w])}{\sum_{h'=1}^H \sum_{w'=1}^W \exp(\mathbf{l}_i^+[h', w'])}, \quad \mathbf{q}_i^-[h, w] = \frac{\exp(\mathbf{l}_i^-[h, w])}{\sum_{h'=1}^H \sum_{w'=1}^W \exp(\mathbf{l}_i^-[h', w'])}.$$

We scale the logits using the softmax maps to increase the contribution of locations that include the class. Combining them results in our final logits

$$\mathbf{p}_i^+ = \sum_{h=1}^H \sum_{w=1}^W \mathbf{q}_i^+[h, w] \cdot \mathbf{l}_i^+[h, w], \quad \mathbf{p}_i^- = \sum_{h=1}^H \sum_{w=1}^W \mathbf{q}_i^-[h, w] \cdot \mathbf{l}_i^-[h, w].$$

The final output is the concatenation of the global positive and negative logits:

$$\mathbf{p}_i = [\mathbf{p}_i^+, \mathbf{p}_i^-] \in \mathbb{R}^{2N}.$$

## C  ALGORITHM AND PSEUDO CODE

---

**Algorithm 1** DCLIP Pipeline

---

**Require:** Multi-label dataset $\mathcal{D} = \{(x_i, y_i)\}_{i=1}^N$ with $x_i$: image, $y_i$: labels
  Pre-trained CLIP image encoder $f_{\theta,\text{img}}$ and text encoder $f_{\theta,\text{text}}$ (frozen)
  Learnable image and text projectors $h_{\phi,\text{img}}, h_{\phi,\text{text}}$
  Positive and negative prompts $\{\text{txt}_j^+, \text{txt}_j^-\}_{j=1}^C$ for each class
1: **for** each training batch **do**
2:   Extract local visual features: $z_i \leftarrow f_{\theta,\text{img}}(x_i)$ (without final pooling)
3:   Encode text prompts: $t_j^+ \leftarrow f_{\theta,\text{text}}(\text{txt}_j^+), \quad t_j^- \leftarrow f_{\theta,\text{text}}(\text{txt}_j^-)$
4:   Project features:

$$z_i' \leftarrow h_{\phi,\text{img}}(z_i), \quad {t'_j}^+ \leftarrow h_{\phi,\text{text}}({t'_j}^+), \quad {t'_j}^- \leftarrow h_{\phi,\text{text}}({t'_j}^-)$$

5:   Concatenate projected text features: $t' = [t'^+, t'^-]$
6:   Compute self-similarity matrix: $S = (t')^\top t'$
7:   Compute **MFI Loss**:

$$\mathcal{L}_{\text{MFI}} = \sum_i (S_{ii} - 1)^2 + \lambda \sum_{i \neq j} S_{ij}^2$$

8:   **for** each location $(h, w)$ in $z_i'$ **do**
9:     Compute positive and negative similarity maps:

$${s_j}^+ = \langle z_i'(h, w), {t'_j}^+ \rangle, \quad {s_j}^-(h, w) = \langle z_i'(h, w), {t'_j}^- \rangle$$

10:   **end for**
11:   Aggregate similarity maps across spatial dimensions to get logits $p_i$
12:   Compute **ASL Loss** between $p_i$ and ground truth $y_i$: $\mathcal{L}_{\text{ASL}}$
13:   Combine losses: $\mathcal{L}_{\text{total}} = \mathcal{L}_{\text{ASL}} + \alpha \mathcal{L}_{\text{MFI}}$
14:   Update projector parameters $\phi$ via gradient descent
15: **end for**
16: **return** Frozen projectors $h_{\phi,\text{img}}, h_{\phi,\text{text}}$

---

**Algorithm 2** Pseudocode for DCLIP

```python
# f: image encoder
# g: image projector
# h: text projector

# encode image and apply projector
image_feat, attn = f(image)                           # [B, D, N]
image_proj = g(image_feat.permute(0, 2, 1))           # [B, N, C]
image_proj = image_proj.permute(0, 2, 1)              # [B, C, N]
image_proj = normalize(image_proj, dim=1)

# encode text prompts
text_pos = encode_text(tokenized_prompts_pos)         # [K, D]
text_neg = encode_text(tokenized_prompts_neg)         # [K, D]
text = torch.cat([text_neg, text_pos], dim=0)         # [2K, D]
text_proj = h(text)                                   # [2K, C]
text_proj = normalize(text_proj, dim=-1)

# MFI Loss
c = bn(text_proj).T @ bn(text_proj)                   # [C, C]
collapse_prevention = ((c.diag() - 1)**2).sum()
mfi_reduction = off_diagonal(c).pow(2).sum()
Loss_MFI = collapse_prevention + λ * mfi_reduction

# class-specific attention map
score = conv1d(image_proj, text_proj[:, :, None]) # [B, 2K, N]
weights = softmax(score, dim=-1)
aggregated = (score * weights).sum(dim=-1) * 5    # [B, 2K]

logits = aggregated.view(B, 2, -1)
Loss_ASL = ASL(logits, ground_truth)

Loss_DCLIP = Loss_ASL + α * Loss_MFI
```

# D ABLATIONS

## D.1 MFI REDUCTION - COLLAPSE PREVENTION ABLATION

We study the effect of $\lambda$, which controls the trade-off in the number of MFI reduction terms and collapse prevention terms in the MFI loss eq. 2. As the self-similarity matrix in eq. 2 includes a larger number of MFI reduction terms compared to the collapse prevention terms, $\lambda$ is introduced to balance their contributions. As shown in Figure 10, the model achieves stable performance across a wide range of $\lambda$ values (0.02 to 0.20), with mAP varying by less than 0.5. This robustness suggests that the method is not overly sensitive to the exact loss balance and can generalize well without extensive tuning of $\lambda$.

## D.2 BCE VS FOCAL VS ASL

Tab. 6 presents the results of combining the proposed Mutual feature information (MFI) loss with standard multi-label recognition losses on the COCO dataset. When combined with Binary Cross-Entropy (BCE), MFI achieves a mAP of 81.2. Incorporating Focal Loss leads to a significant improvement, reaching 83.8 mAP. The best performance is obtained by combining MFI with Asymmetric Loss (ASL), achieving 85.6 mAP.

Table 6: **Multi-label Losses Ablation** Performance on COCO using the proposed MFI loss combined with different multi-label losses.

| MFI | mAP (COCO) |
|-----|-----------|
| + BCE | 81.2 |
| + Focal | 83.8 |
| + ASL | 85.6 |

## D.3 POOLING-PROJECTION ABLATION

We study the effect of projection order by comparing two variants: (1) applying global pooling before projection (*Pooling → Projection*) and (2) projecting local features first and then aggregating them with softmax attention (*Projection → Pooling*). As shown in Tab. 7, pooling before projection leads to a clear performance drop (81.3 mAP) compared to our design (85.6 mAP). This confirms that projecting local features prior to pooling is crucial, since it preserves spatial information and allows the softmax attention to focus on discriminative regions before aggregation.

Table 7: **Pooling–Projection ablation on COCO.** Preserving local features by projecting before pooling yields substantially better performance.

| Method | mAP (COCO) |
|--------|-----------|
| Pooling → Projection | 81.3 |
| Projection → Pooling | 85.6 |

## D.4 PERFORMANCE ON CLIP ARCHITECTURES

In Tab. 8 we evaluate DCLIP on ResNet 50 and ResNet 101, ViT-B/16 and Vit-B/32. For ViT-based models, we adapt our projectors to handle different feature dimensions while maintaining the same architectural principles. The image projector processes features from ViT's final layer patches, without using the CLS token. Hyperparameters $\alpha$ and $\lambda$ remain unchanged across architectures.

# E MAP VS MIOU

Figure 11 presents a comparison between multi-label recognition (mAP) and zero-shot semantic segmentation (mIoU) performance across VOC 2012, COCO with background, and COCO without

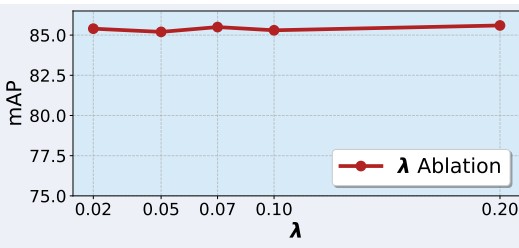

Figure 10: **Effect of** ($\lambda$) **on mAP.** $\lambda$ controls the trade-off between MFI reduction and collapse prevention terms in MFI loss. Performance remains stable across a range of $\lambda$ values, indicating robustness to the choice of this hyperparameter.

Table 8: **DCLIP Performance Across CLIP Architectures.** We evaluate DCLIP across RN-50, RN-101, ViT-B/16, and ViT-B/32.

| DCLIP Backbone | Param (M) | VOC | | COCO | |
| --- | --- | --- | --- | --- | --- |
| | | GPU hrs | mAP | GPU hrs | mAP |
| RN50 | 1.2 | 3 | 94.4 | 13 | 83.2 |
| RN101 | 0.4 | 3 | 95.4 | 13 | 85.6 |
| ViT-B/16 | 0.4 | 3 | 94.6 | 13 | 84.9 |
| ViT-B/32 | 0.4 | 3 | 94.6 | 13 | 84.4 |

background settings. We observe a consistent positive correlation between mAP and mIoU across all configurations, suggesting that improvements in multi-label classification translate to better zero-shot segmentation. Notably, VOC 2012 and COCO (No Bkgd) exhibit stronger segmentation performance compared to COCO (Bkgd) at similar mAP levels, highlighting the challenge introduced by background classes in segmentation tasks.

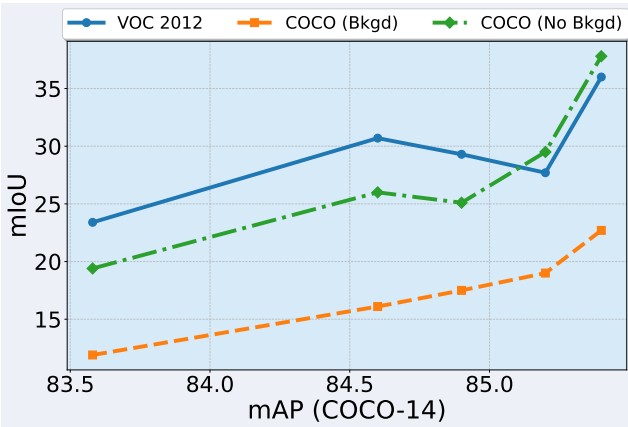

Figure 11: mAP vs mIoU. Performance comparison of zero-shot semantic segmentation (mIoU) for VOC2012, COCO 2017 with and without the background, and VOC Context as a function of multi-label recognition (mAP) performance on the COCO-14 dataset. A general trend: higher MLR performance positively correlates with segmentation results.

