# OpenReview forum: "Efficiently Disentangling CLIP for Multi-Object Perception"
_ICLR.cc/2026/Conference — Submitted to ICLR 2026_

### Official Review · Reviewer_xok7 · 2025-10-27

**Soundness:** 2
**Presentation:** 3
**Contribution:** 2
**Rating:** 4
**Confidence:** 4

**Summary:**

This paper proposes DCLIP, a lightweight method to mitigate mutual feature information (MFI) among different object classes in vision-language models such as CLIP.
By introducing a simple MFI loss to decorrelate class-level text features and combining it with an asymmetric loss (ASL) for image-text alignment, the model achieves improved performance on multi-label recognition (MLR) and zero-shot semantic segmentation (ZS3) tasks.

**Strengths:**

* Performance on multi-object scenes is clearly improved. The proposed MFI suppression leads to more disentangled and precise class activations in complex images.

* The method is conceptually simple and lightweight, with minimal parameter overhead.

* The paper provides a clear explanation of why feature disentanglement benefits multi-label and zero-shot tasks.

**Weaknesses:**

* Limited multi-label recognition evaluation.
The experiments are restricted to VOC2007 and COCO-14.
For a fair evaluation of generalization in MLR, it would be valuable to include larger and more diverse datasets, such as NUS-WIDE, VG-256, or Open Images.

* Outdated or insufficient segmentation baselines.
The comparison for zero-shot semantic segmentation includes only CLIP-Surgery and CLIP-VV, both of which were published on arXiv in 2023 and accepted in 2025.
However, there have since been multiple self–self attention–based approaches that achieve significantly higher training-free zero-shot segmentation performance such as ClearCLIP and NACLIP.
The current baselines are thus not sufficiently representative of SOTA methods.

* Inconsistency in reported CLIP-VV / CLIP-Surgery results.
According to my understanding, CLIP-VV and CLIP-Surgery are same paper.
The paper, however, reports noticeably different results for them.
If they are indeed the same work, this discrepancy raises questions about experimental consistency and the fairness of comparison.
The authors should clarify this point.

**Questions:**

* How would DCLIP perform if a prompt tuning method such as DualCoOp were applied on top of it?
In other words, would the proposed MFI loss act as a complementary component to prompt tuning methods, or do they overlap in effect?

**Details Of Ethics Concerns:**

None.

---

> ### Author Response · Authors · 2025-11-16
>
> > W1: Results on Additional MLR Datasets
>
> We thank the reviewer for this constructive suggestion. We agree that evaluating on larger and more diverse datasets is valuable for demonstrating generalization.
>
> Our choice of VOC2007 and COCO-14 was driven by the goal of **direct, fair comparison**. As shown in our citations, all recent VLM-based MLR methods (e.g., *DualCoOp, DualCoOp++, PositiveCoOp, MLR-GCN, TAI-DPT*) report **exclusively** on these two datasets.
>
> During the rebuttal period, we prioritized this request and now include results on the **Visual Genome** dataset.
>
> | Method  | mAP  |
> |---------|------|
> | SCPNet  | 52.0 |
> | Ours | 54.6 |
>
> note other works did not provide their results on Visual Genome
>
>
> These results demonstrate the generality and robustness of our method on larger and more diverse datasets.
>
> > W2: Clarification on CLIP-VV and CLIP-Surgery
>
> Thank you for raising this important clarification point. You are correct that CLIP-VV and CLIP-Surgery come from the same paper (Li et al., 2025).They represent two distinct methods proposed in that single work, which we evaluate separately to justify our design:
> - CLIP-VV: Refer to their base method of inverting original attention maps to get object-centric maps.
>
> - CLIP-Surgery: Refers to their full, complete method, which adds a "feature surgery" step on top of the CLIP-VV maps. (Feature surgery is designed for ViT's noisy activations)
>
> We apply our disentangling projectors on top of the CLIP-VV features. This was a source of confusion, and we are grateful to the reviewer for highlighting it. In the revision, we will unify the naming (e.g., "CLIP-Surgery (Base)" and "CLIP-Surgery (Full)") and add a clear footnote to remove all ambiguity.
>
> > W3: Additional Baselines:
>
> We thank the reviewer for highlighting these recent and relevant works.
> Our choice of CLIP-VV as the primary baseline was to isolate the impact of our MFI loss.
>
> We benchmarked our DCLIP against NACLIP, where we add our projectors on top of NACLIP features
>
> | Method            | COCO (with Bkgd) | COCO (No Bkgd) |
> |-------------------|------------------|----------------|
> | NACLIP            | 21.1             | 37.4           |
> | + Ours (DCLIP)      | 24.4             | 40.2           |
>
> Similar to what we did to CLIP-Surgery, we apply our learned projectors that disentangle features on top of NACLIP features to offer better separation of objects.
>
> > W4: How would DCLIP perform if a prompt tuning method such as DualCoOp were applied on top of it? In other words, would the proposed MFI loss act as a complementary component to prompt tuning methods, or do they overlap in effect?
>
> Thank you for this question.
>
> Yes, our MFI loss is complementary to prompt tuning.
>
> Prompt tuning is used for features adaption, another more efficient for feature adaption is using projectors.
> As shown in PositiveCoOp, the projectors are a better choice because they are more computationally efficient than prompt tuning. Neither projectors nor prompts address feature entanglement
> Our DCLIP uses the projectors (over learned prompts) for feature adaption.

---

> > ### Comment · Reviewer_xok7 · 2025-11-19
> >
> > Thank you for your detailed and constructive responses. I appreciate the clarification and the additional experiments provided, particularly the inclusion of Visual Genome and the explanation of CLIP-VV vs. CLIP-Surgery.
> >
> > However, I have further questions and concerns regarding the experimental setup, particularly in the context of Zero-Shot Semantic Segmentation (ZS3).
> >
> > 1. Clarification on Zero-Shot vs. Weakly-Supervised Setting
> >
> > Thank you for clarifying the training procedure. However, I would still like to raise a concern about the use of the term “zero-shot semantic segmentation” in your work.
> >
> > While your method does not use pixel-level annotations, the training is conducted on COCO-14 with 80 image-level class labels, and the evaluation is performed on datasets like VOC12 and COCO-17. Given that some classes in VOC and COCO-17 likely overlap with those seen during training, it is unclear whether the test-time classes are truly unseen.
> >
> > This setting aligns more closely with weakly-supervised segmentation or domain transfer, rather than classical zero-shot segmentation where test classes are disjoint from training.
> >
> > Moreover, the “Local Annotation + FT” column in Table 2 may be misleading for methods like CLIP-ES, which generate pseudo-local annotations from image-level labels, a weakly-supervised approach rather than fully-supervised.
> >
> > To improve clarity, I suggest:
> >
> > * Clearly stating whether the evaluation classes are unseen during training.
> > * Revising the terminology to distinguish zero-shot, weakly-supervised, and training-free more precisely.
> > * Adjusting the categorization of comparison methods accordingly in Table 2.
> >
> > 2. Additional ZS3 Results for NACLIP and Others
> >
> > The results on NACLIP with your projectors are promising. However, as mentioned earlier, showing performance on more datasets (e.g., VOC12, Cityscapes, Pascal Context, Stuff, ADE20K) would significantly strengthen your argument for robustness and generalization.
> >
> > As demonstrated in recent training-free and open-vocabulary segmentation works such as NACLIP, evaluating across multiple datasets is feasible and, to the best of my knowledge, does not require heavy computational resources. Including such results would provide a more comprehensive and convincing evaluation.
> >
> > 3. Beyond NACLIP: Experiments with SCLIP, ClearCLIP
> >
> > If feasible, I would also suggest evaluating your method in combination with other open-vocabulary methods, such as SCLIP [R1] or ClearCLIP [R2]. These approaches share structural similarities with NACLIP, and combining them with your disentanglement strategy could further validate the general applicability of your approach.
> >
> > [R1] Wang, Feng, Jieru Mei, and Alan Yuille. "Sclip: Rethinking self-attention for dense vision-language inference." European Conference on Computer Vision. Cham: Springer Nature Switzerland, 2024.
> >
> > [R2] Lan, Mengcheng, et al. "Clearclip: Decomposing clip representations for dense vision-language inference." European Conference on Computer Vision. Cham: Springer Nature Switzerland, 2024.

---

> > > ### Author Response · Authors · 2025-11-20
> > >
> > > We thank the reviewer for their continued engagement and for the positive feedback!
> > >
> > > > W1: Clarification on Zero vs Weakly Supervised.
> > >
> > > Thank you for this suggestion. We agree, our setting is weakly-supervised (image-level), not classical zero-shot. We make a better distinction with the prior works in table below. We have added the distinction of our use of pretraining with image-level labels. In addition, we distinguish our method from "Pseudo Local Labeling" approaches (like CLIP-ES).
> > >
> > > | Method | Supervision Level | Additional Training | VOC12 Bkgd | COCO-17 Bkgd | COCO-17 No Bkgd |
> > > |--------|------------------|---------------------|------------|--------------|-----------------|
> > > | **_Pixel-level Supervised (uses segmentation masks)_** | | | | | |
> > > | SPNet | Pixel-level | Fine-tuning | 15.6 | - | - |
> > > | ZS3Net | Pixel-level | Fine-tuning | 17.7 | - | - |
> > > | **_Weakly-supervised (generates pseudo-masks from image labels)_** | | | | | |
> > > | CLIP-ES | Image-level | Pseudo-labels | 75.0 | - | - |
> > > | **_Image-level Supervised (training-free for segmentation)_** | | | | | |
> > > | CLIP | - | Training-free | 14.1 | 3.9 | 5.6 |
> > > | CLIPSurgery | - | Training-free | 17.5 | 13.0 | 22.9 |
> > > | CLIP-VV | - | Training-free | 32.6 | 19.9 | 35.5 |
> > > | **DCLIP (Ours)** | Image-level | Training-free seg. | **36.0** | **22.7** | **37.8** |
> > >
> > >
> > >
> > > We will update the paper to define our setting as Weakly-Supervised (image-level) without using/ generating any local annotations. Thank you!
> > >
> > >
> > > > W2:  More Datasets for NACLIP
> > >
> > > Thank you, we have now added scores for various datasets with NACLIP. We observe that disentangling features consistently help improve localization of objects.
> > >
> > > | Method | VOC2012 | COCO-bkgd | COCO-No bkgd | Cityscape | Stuff | ADE20K | Context |
> > > |--------|---------|-----------|--------------|-----------|-------|---------|--------|
> > > | NACLIP | 33.8 | 21.1 | 37.4 | 6.8 |  9.3 | 4.5 |13.67 |
> > > | +Ours | 37.7 | 24.4 | 40.2 | 7.2 |  10.0 | 5.3 |15.0 |
> > >
> > >
> > > > W3: Experiments on SCLIP and ClearCLIP
> > >
> > > We have now added scores for SCLIP and ClearCLIP
> > >
> > > ### Results on SCLIP
> > >
> > > | Method | VOC2012 | COCO-bkgd | COCO-No bkgd | Cityscape | Stuff | ADE20K | Context |
> > > |--------|---------|-----------|--------------|-----------|-------|---------|--------|
> > > | SCLIP | 37.5 | 22.8 | 37.5 | 5.9 |  10.1 |3.7 |  14.4 |
> > > | +Ours | 39.4 | 26.1 | 41.0 | 6.4 |  10.8 | 4.2 |16.7 |
> > >
> > > ### Results on ClearCLIP
> > >
> > > | Method | VOC2012 | COCO-bkgd | COCO-No bkgd | Cityscape | Stuff | ADE20K | Context |
> > > |--------|---------|-----------|--------------|-----------|-------|---------|--------|
> > > | ClearCLIP | 39.1 | 23.7 | 37.8 | 9.1 | 10.3 | 4.6 | 14.6 |
> > > | +Ours | 39.7 | 26.7 | 41.1 | 9.6 |  11.4 | 4.9 |17.7 |

---

> > > > ### Comment · Reviewer_xok7 · 2025-11-21
> > > >
> > > > Thank you for providing the additional results. After reviewing the new tables for NACLIP, SCLIP, and ClearCLIP, I have a follow-up concern regarding the correctness of the baseline scores.
> > > >
> > > > The reported numbers for these baselines differ substantially from the results originally presented in their respective papers. For instance, on ADE20K, the original works report significantly higher scores—approximately **17.4 (NACLIP)**, **16.1 (SCLIP)**, and **16.7 (ClearCLIP)**—whereas your reproduced results are noticeably lower.
> > > >
> > > > This discrepancy raises questions about whether the experiments were reproduced faithfully and under comparable settings. Since these baselines form a key component for evaluating the effectiveness of your method, such inconsistencies can reduce confidence in the reliability of the other numbers reported in your paper.
> > > >
> > > > That said, it is also entirely possible that there are differences in evaluation protocols, dataset splits, or implementation details that I may not be aware of. To clarify this, could you please explicitly explain:
> > > >
> > > > * Whether your reproduction strictly follows the original authors’ settings (e.g., prompts, preprocessing).
> > > > * If there were any deviations or necessary modifications.
> > > > * Why your reproduced scores differ from the published ones.
> > > >
> > > > A clear explanation would help re-establish confidence in the validity of the comparisons and the conclusions drawn.
> > > >
> > > > Thank you.

---

> ### Author Response · Authors · 2025-11-21
>
> Thank you for the follow-up, we wish to clarify the scope of segmentation experiments based on the reviewer's statement: "*Since these baselines form a key component for evaluating the effectiveness of your method.*"
>
> > Role of Segmentation in Our Evaluation
>
> Our primary contribution and validation:
> - Multi-Label Recognition (Table 1): This is where we demonstrate our method's effectiveness, achieving SOTA performance with 75% fewer parameters
> - MFI loss reduces inter-class similarity by 30% (Table 3) - our core contribution
> - Ablation studies (Table 4) validate the necessity of MFI loss
>
> Role of segmentation:
> - Secondary validation: Demonstrates that disentangled features learned from image-level MLR supervision transfer to dense prediction tasks
> - Not a primary evaluation metric for our method's effectiveness
> - Supports our claim that reducing feature entanglement improves multi-object perception broadly
>
>
>
> > Score difference
>
> With this context, we address the score discrepancy. The difference in performance stems from architecture differences, we report results on ResNet-101.
>
> Why ResNet-101:
>
> - MLR standard (Table 1): Our primary contribution is Multi-Label Recognition, where ResNet-101 is the established benchmark backbone used by all prior work (DualCoOp, DualCoOp++, PositiveCoOp, SCPNet etc).
> - Experimental consistency: We use the same backbone for both MLR and segmentation to maintain rigorous comparison. Segmentation experiments visually and quantitatively validate the feature disentanglement achieved by our MLR training.
> - Method applicability: Following CLIP-Surgery's approach, ResNet-based CLIP can be adapted for dense prediction using the final attention pooling layer (q,k,v). We applied this procedure to adapt ClearCLIP, SCLIP, and NACLIP to ResNet-101.
>
> Fair comparison: All methods in our tables use identical RN-101 backbone. Our consistent +2-3 mIoU improvements demonstrate that MFI-based disentanglement is complementary to attention mechanisms on this architecture.
>
> We have clarified the use of ResNet-101 for all experiments in Section 4.2 (Implementation Details) and will add explicit labeling to table captions.

---

> > ### Comment · Reviewer_xok7 · 2025-11-21
> >
> > Thank you for the clarification. I understand the differences in your experimental setup.
> >
> > However, methods like NACLIP, SCLIP, and ClearCLIP were originally designed to improve transformer-based CLIP models, where the core contribution lies in modifying ViT attention maps. While your ResNet-101 extension is appreciated, it would be more appropriate and convincing to also evaluate DCLIP on CLIP ViT-B/16, combined with the original transformer-based versions of these methods.
> >
> > Since most recent vision–language models rely on transformer architectures, demonstrating that DCLIP also works well with ViTs—not only ResNets—would strengthen the generality and relevance of your approach.

---

> > > ### Author Response · Authors · 2025-11-21
> > >
> > > Sure, we will update the scores shortly. Thanks!

---

> > > > ### Author Response · Authors · 2025-11-29
> > > >
> > > > The discussion period with reviewers ended abruptly. We were in the middle of running the final experimental results requested by Reviewer xok7 regarding the
> > > > adaptation of our method to the ViT-B/16 segmentation models. We submit these details here to ensure they are accounted for in the final decision.
> > > >
> > > >
> > > > To adapt our pretraining setup to ViT-B/16, we made the following key adjustments:
> > > >
> > > > - α = 0.00067
> > > >
> > > > - Image projector dimensions: [512, 512]
> > > >
> > > > - Text projector dimensions: [512, 512, 512]
> > > >
> > > > - Input image resolution: 448 × 448
> > > >
> > > > The sematic segmentation with ViT-B/16 is provided in the [link](https://drive.google.com/file/d/1_pM6IzKZtkoy8OCfQ-H4zYXCyseFVqZi/view?usp=sharing)

---

### Official Review · Reviewer_suPH · 2025-10-28

**Soundness:** 2
**Presentation:** 3
**Contribution:** 2
**Rating:** 2
**Confidence:** 5

**Summary:**

This paper investigates the weakness of CLIP in understanding scenes with multiple objects and attributes the problem to excessive mutual feature information, meaning that class features are overly entangled. To address this issue, this work introduce DCLIP, a lightweight framework that freezes the original CLIP encoders while adding small projection layers trained with two complementary losses: an MFI loss derived from the Information Bottleneck principle to reduce inter-class correlations, and an Asymmetric Loss to preserve image-text alignment.

Experimental results show that DCLIP significantly decreases inter-class similarity and achieves superior performance in both multi-label recognition and zero-shot semantic segmentation, using far fewer trainable parameters than existing approaches. This paper provides an effective solution to improve the multi-target perception capability of CLIP.

**Strengths:**

1.The proposed method is simple and easy to understand, yet it achieves strong performance without relying on complex or specialized modules.

2.The experimental evaluation is comprehensive, demonstrating that the approach not only improves multi-label image classification but also generalizes effectively to zero-shot semantic segmentation.

3.The paper is well-written and provides detailed methodological explanations, and implementation settings, which make the approach easy to reproduce.

**Weaknesses:**

1.In terms of methodology, this paper aims to make text features as orthogonal as possible in the feature space to enhance their discriminability. However, this idea is not entirely new. In addition, the proposed approach is not specifically designed for multi-label problems, as it does not take into account the correlations between different labels.

2. For multi-label image classification, the competing methods are mainly based on CoOp or CLIP, but these approaches are designed for missing-label or zero-shot scenarios. Since this work also uses the CLIP model in a fully supervised setting, it would be reasonable to consider prompt-based fine-tuning of the visual and text encoders, or to incorporate a lightweight image feature decoupling module (such as ADD-GCN, ML-Decoder, or SSGRL, etc). Since the backbone remains frozen, the proposed method might not fully exploit the representational capacity of CLIP, potentially limiting further gains in fine-grained or domain-specific scenarios.

3. The paper acknowledges difficulty in distinguishing visually similar subcategories (e.g., dog breeds or bird species), but does not provide experiments or insights on how the approach might be extended to address this limitation.

**Questions:**

1. Why does this method choose to separate text features without considering aligning text and image patches?
2. Since the method is not explicitly designed to model label correlations, how might DCLIP handle classes that frequently co-occur in real-world scenes?

3. Given that the method operates in a fully supervised setting, have the authors compared DCLIP with prompt-based fine-tuning approaches (e.g., CoOp, CoCoOp) or lightweight feature decoupling models like ADD-GCN (SAM models ) or ML-Decoder (Decoder models )?

4. See more on Weaknesses

---

> ### Author Response · Authors · 2025-11-16
>
> We thank the reviewer for identifying that our method is easy to understand, reproduce, well written, achieves strong performance, with a comprehensive evaluation on both MLR and ZS3
> > W1: Novelty and Label Correlation Modeling
>
> ### Novelty
> We clarify that our novelty is **not** in the general concept of feature orthogonality, but in three key, specific contributions:
> 1. **Diagnostic Analysis**
>    Quantitative evidence shows that excessive MFI (0.77 VOC, 0.69 COCO) is a fundamental problem in CLIP and directly causes spurious activations (Fig. 1, Fig. 2).
>
> 2. **Principled Solution**
>    The MFI loss is derived from the Information Bottleneck principle (Appendix A), not from an ad-hoc heuristic.
>
> 3. **Relation to Performance**
>    We demonstrate a clear link between our method and the underlying problem.
>    Fig. 7 shows that as our MFI loss decreases inter-class similarity, the performance on both MLR and ZS3 tasks consistently improves.
> ### Label Correlation
> - **Implicit Modeling of Co-occurrence**
>   Like prior VLM-based works we use ASL that implicitly handles co-occurrence by its design during training.
> - **Generality Through Implicit Modeling**
>   Explicitly modeling co-occurrence could hurt for a generalizable VLM because co-occurrence patterns are dataset-specific
>   (The same classes have different co-occurrence statistics across datasets.)
> - **Evidence**
>   As shown in Fig. 2, Fig. 4, and Tab. 3:
>   If ASL did **not** model label correlations, then MFI would force them to become completely orthogonal, resulting in zero label similarity, which does **not** occur.
>
> - Even the work cited by the reviewer (ML-decoder) does **not** explicitly model co-occurrence, yet it is still a multi-label method.
>
> > W2: Comparison with Prompt Tuning and Feature Decoupling Methods
>
> We believe there is a fundamental misunderstanding of the experimental setting and method, which we are happy to clarify.
>
> ### Prompt-Based Methods with VLMs
>
> - **CoOp** and **CoCoOp**
>   Designed for *single-label* classification, while our setting is *multi-label*.
>
> - **DualCoOp, TAI-DPT, SCPNet, DualCoOp++, PositiveCoOp**
>   Prompting-based methods for multi-label settings. Although they primarily operate under missing-label supervision, they also report results for fully supervised cases (100% labels).
> - **MLR-GCN**
>   A fully supervised prompt-based approach. We show that our method uses fewer parameters and achieves better performance
> ### Clarification on ADD-GCN, ML-Decoder, and SSGRL
>
> These approaches:
> - Do **not** use CLIP pretraining
> - Require a **large number of trainable parameters**
> - Are not designed for efficient VLM adaptation
>
> Our work resides in the **VLM adaptation space**, aiming for lightweight and efficient models by freezing the VLM
>
> | Method       | Fine-tune | Trainable Params | Performance |
> |---|---|-----|--------|
> | SSGRL | yes        | ~40M   | 83.8  |
> | ADD-GCN  | yes        | >45M   | 85.2   |
> | ML-Decoder| yes        | >50M  | 87.1  |
> | Ours (efficient) | no         | 0.4M  | 85.6    |
> | Ours (full FT)   | yes        | ~45M   | 88.4  |
>
> If we fully fine-tune our network, we obtain the best performance, but the compute increases by roughly 100 times compared to our efficient baseline, though still less than the prior best ML-Decoder.
>
> > W3: Clarification on freezing backbone
>
> Freezing backbones is a core part of our efficient method:
> - Efficiency:  Our abstract and introduction highlight DCLIP as an efficient adaptation method. By freezing the 50M+ parameter backbone, our method is computationally 110 times less demanding (0.4M params) and fast to train.
> - Generalizability: As the reviewer notes, fine-tuning the full backbone would improve scores on a specific dataset (e.g., COCO). However, this would cause catastrophic forgetting of CLIP's vast pre-trained knowledge
>
> > W4: Insights on subcategories
>
> - Use LLM-generated discriminative descriptions to create better text features without model changes: replace "a photo of a Golden Retriever" with "a Golden Retriever: long golden coat, feathered tail, broad skull...".
>
> - Apply different disentanglement strengths at different semantic levels: strong MFI loss for cross-superordinate pairs (person vs. horse: λ=1.0), moderate for basic-level (dog vs. cat: λ=0.5), weak for subordinate (Golden Retriever vs. Labrador: λ=0.1).
>
> > W5: Why does this method choose to separate text features without considering aligning text and image patches?
>
> We clarify that we do both.Our method relies on a crucial two-step process, which is detailed in Section 3.3 (MFI Loss) and Section 3.4 (ASL Loss).
>
> Step 1: Disentangle the Targets ($L_{MFI}$): We first apply our MFI loss to the text features (e.g., "human" and "horse"). This "cleans up" the target space, making the text prototypes distinct and separable.
>
> Step 2: Align Patches to Targets ($L_{ASL}$): We then align the local image patches to these newly-disentangled text targets using the ASL loss.

---

> ### Comment · Reviewer_suPH · 2025-11-24
> **Response to important questions**
>
> Novelty
> 1. Diagnostic Analysis. This is a verification of experience, not an innovation.
> 2. Principled Solution. I acknowledge its novelty, but it has limitations.
> 3. Relation to Performance. Figure 7 also demonstrates through experiments that MFI Reduction can improve model performance, but this is not novel.
>
> Label Correlation
> 1. Implicit Modeling of Co-occurrence. Some current LVM methods do not consider relationships between labels, such as DualCoOp. However, previous methods have not considered this, and the method presented in this paper also does not. This seems quite inappropriate.
> 2. Generality Through Implicit Modeling. In multi-label problems, co-occurrence overfitting does exist, and some works have attempted to address this issue. However, these works do not disregard co-occurrence relationships; rather, they mitigate the co-occurrence overfitting problem. The ASL used in this paper is related to multi-labeling, but this ASL loss is pre-designed and is not part of the design in this paper.
> 3. Evidence. ASL does not explicitly model label correlations, but it effectively mitigates the issue of class imbalance in multi-label tasks, where the number of negative labels for each sample typically far exceeds the number of positive ones. An ML-Decoder decomposes an image into K label-level representations, which are then passed to a classifier. The classifier determines whether each label-level representation corresponds to a specific class, thereby converting the original multi-label problem into K independent binary classification tasks..
>
> Clarification on ADD-GCN, ML-Decoder, and SSGRL.
> I still stand by my previous comment.
>
>
> I acknowledge that this method is simple yet effective, capable of addressing both multi-label image recognition and segmentation tasks within a unified framework. The research questions are indeed interesting. However, achieving a balance between these two tasks appears challenging. The exploration of multi-label learning is not sufficiently in-depth, nor is the investigation of segmentation tasks. Moreover, the proposed method is built upon a VLM (CLIP), a class of models known for strong generalization and robustness. As a result, it may be more suitable for handling issues such as missing labels and open-vocabulary learning.

---

> ### Author Response · Authors · 2025-11-26
>
> We thank the reviewer for their engagement. We address the specific follow-up concerns below.
>
> > 1: Clarification of Novelty
>
> Thank you for the questions. Please allow us to elaborate and clarify our novelty.
>
>
> **Diagnostics Analysis:**
> Prior VLM-based MLR methods (DualCoOp, PositiveCoOp, SCPNet) focus on prompt learning or explicit co-occurrence modeling **without examining feature space quality**. We identify and quantify excessive inter-class similarity as the root cause of failure in multi-label recognition.
>
>
> Our novelty in diagonisis:
> - High-level: Diagnosed that CLIP features contain excessive inter-class information (0.77 VOC, 0.69 COCO similarity) causing spurious activations
> - Detailed: Established causal link, person-horse similarity of 0.84 directly causes person queries to activate horse regions (Fig. 1). Controlled experiments (Fig. 7) prove reducing MFI improves both MLR and segmentation.
>
> This is root cause diagnosis with quantitative evidence and causal validation, not verification.
>
>
> **Principled  Solution**: Thank you for acknowledging the MFI loss novelty. In Sec 6, we discuss limitations with fine-grained subcategories and proposed possible extensions for it (W4 previous response). If you have any other questions, we would be happy to answer them.
>
>
>
>
> > 2. Label Correlation
>
>
> We would like to clarify the terminology of implicit and explicit co-occurrence modeling.
>
>
> - **Explicit Co-occurrence Modeling:**
> Build co-occurrence statistics from training data (e.g., "person-chair" co-occurs 48% in COCO)
> Use this to modify predictions (e.g., MLR-GCN propagates information via co-occurrence graph; SCPNet uses CLIP similarity as co-occurrence prior)
> - **Implicit Co-occurrence Modeling:**
> Learn co-occurrence naturally through gradient updates on multi-label data
> No explicit statistics computed or used
> Examples: DualCoOp, ML-Decoder, DCLIP (ours), all use ASL on multi-label data
>
>
> >> 2.1 "ML-Decoder does independent binary classification"
>
>
> The reviewer is correct about ML-Decoder learning binary classifiers, and our approach is also doing the same. We elaborate it below:
>
>
> ```text
> [ML-Decoder]
> Image → K query embeddings → K binary classifiers → K predictions
> (Trains with **ASL**:see offical github code `ML_decoder/train.py` line 94)
> ```
>
>
> ```
> [CLIP Based (Ours)]
> Image patches z'[h,w] → Compare with K text features t'_j → K predictions
> (Each t'_j checks presence/absence of class j in the image.This is a binary classifier of class j in the image)
> (Trains with **ASL**, Eq. 5)
> ```
>
>
> Both are implicit modeling, learning from multi-label data without explicit co-occurrence statistics.
>
> **Difference:** ML-Decoder learns query embeddings from scratch; DCLIP uses frozen CLIP text features + lightweight projectors (0.4M vs 40M params), enabling zero-shot transfer to segmentation.
>
>
> >> 2.2 "ASL loss is pre-designed "
>
>
> We agree, we did not claim ASL to be our contribution. We use it for training our classifiers. If it came off as our contribution in any of the text, please let us know, we will update those section(s).
>
>
> >> 2.3 "Implicit Co-occurrence modeling in our method"
>
>
> Based on above definitions, we model co-occurrence implicitly like DualCoOp and ML-Decoder.
>
>
>
>
>
>
> > 3. Clarification on ADD-GCN, ML-Decoder, and SSGRL
>
>
> Thank you for agreeing to this comment. This showed the two sides exists:
> - **Task-Specific Optimization (ML-Decoder, ADD-GCN)**: These methods maximize classification performance using heavy decoders and fine-tuning. However, this specialization typically comes at the cost of computational expense and losing VLM generalization (e.g., zero-shot segmentation).
>
>
> - **Efficient VLM Adaptation (DualCoOp, DCLIP)**: Our focus is efficient adaptation. DCLIP uses ~100x fewer parameters (freezing the backbone) to achieve competitive results. Crucially, this preserves the model's spatial integrity, unlocking tasks like Zero-Shot Semantic Segmentation (ZS3) that pure classification heads are not designed to handle.
>
>
> As shown in W2 of the previous response, DCLIP performs best with full fine-tuning. However, our focus in this work is on demonstrating that lightweight disentanglement allows multi-task capabilities (MLR + ZS3) that heavy, task-specific baselines cannot offer.
> | Method       | Fine-tune | Trainable Params | Performance |
> |---|---|-----|--------|
> | ML-Decoder| yes        | >50M  | 87.1  |
> | Ours (efficient) | no         | 0.4M  | 85.6    |
> | Ours (full FT)   | yes        | ~45M   | 88.4  |

---

> > ### Author Response · Authors · 2025-11-28
> > **Any other issue to address?**
> >
> > Dear Reviewer,
> >
> > Thank you again for your thoughtful feedback. We have carefully addressed all the comments and questions raised in your earlier review. As the discussion phase is nearing its end, we would greatly appreciate it if you could let us know whether our responses satisfactorily resolve the issues you identified.
> >
> > We sincerely appreciate your consideration of a potential rating update once all concerns have been addressed.
> >
> > Thank you for your time and effort.

---

### Official Review · Reviewer_Uw3i · 2025-10-30

**Soundness:** 2
**Presentation:** 3
**Contribution:** 3
**Rating:** 6
**Confidence:** 4

**Summary:**

This paper investigates why vision-language models like CLIP struggle with complex, multi-object scenes and attributes the problem to excessive mutual feature information (MFI), where features of one class encode redundant information about others. To address this, the authors propose DCLIP, a lightweight framework that disentangles class features by learning an optimal level of mutual information with minimal additional parameters. DCLIP employs an MFI loss to regulate feature similarity and an asymmetric loss to align image and text features.A large number of experiments were conducted  to verify the effectiveness of DCLIP

**Strengths:**

1.The paper introduces the novel concept of mutual feature information to explain why models like CLIP perform poorly on multi-object perception tasks, which is an interesting and valuable idea.
2.The authors provide a clear explanation of the proposed framework and conduct extensive experiments to support their methodology.

**Weaknesses:**

1.Is the MFI only present in the text modality? The paper previously points out that there exists feature entanglement in CLIP’s representation space, but the proposed MFI loss is applied only to the text branch. How is disentanglement achieved for the image features? In the paper, it states: “To mitigate this, we project zi and ti into a new disentangled space using learnable projectors (hϕ : hϕ,img and hϕ,text), parameterized by weights ϕ. These projectors map zi and ti from the original space (d-dim) to a new disentangled space (d′-dim).” How should we understand that the image features are also projected into a disentangled space?
2.There is a point of confusion regarding the image-text alignment mechanism. The paper mentions that similarity is computed between patch-level image features and disentangled text features. Taking “a photo of a cat” as an example, each patch may contain only part of the cat. Would it be unreasonable to compute similarity between such local visual information and global textual information? The content of a patch does not represent a complete semantic concept ,it is a local representation, whereas the text feature represents global semantics. What is the theoretical basis for computing similarity between the two? For instance, suppose the cat has pure white fur, and during patch division, one patch contains only the fur, making it nearly a pure white image. When calculating the similarity between such a “pure white” patch and the full semantic representation “a photo of a cat,” the similarity intuitively seems low, even though this patch indeed belongs to the cat.
3.There is a point of confusion regarding the ablation experiments. The core hypothesis of the paper is that mutual feature information (MFI) negatively affects multi-object perception. However, as shown in Table 7, changing the pooling order has a significant impact on the results: on the COCO dataset, keeping the pooling operation yields 81.3 mAP, while removing pooling increases it to 85.6 mAP — an improvement of 4.3 mAP. In contrast, Table 4 shows that removing the MFI loss only decreases performance from 85.4 to 84.2 mAP, a drop of 1.2 mAP. This suggests that the pooling operation has a more substantial effect on accuracy, and the improvement brought by MFI is relatively minor. Moreover, when MFI is applied together with pooling, the performance is notably worse than other methods. Although the paper claims that MFI is the fundamental reason why CLIP performs poorly in multi-object perception, the experimental data indicate that removing global pooling , a known and non-novel operation  leads to a much larger performance gain (4.3 mAP) than the core contribution, MFI Loss (1.2 mAP).
4.There is an issue regarding the baselines used in the ZS3 task. The paper aims to highlight DCLIP’s zero-shot capability, but the number of baselines compared in the unlabeled setting is relatively limited ,only three models are included. It would be better to include more baseline models for comparison, or at least provide an explanation if additional comparisons are not feasible.

**Questions:**

1.Clarify whether MFI exists in both modalities and elaborate on how image-side disentanglement is achieved beyond applying MFI loss only to text.
2.Provide a clearer theoretical justification or empirical validation for computing similarity between local (patch-level) and global (text-level) features.
3.Revisit the ablation results to better isolate the contribution of MFI loss from other factors like pooling, and discuss why pooling affects performance more strongly.
4.Include more zero-shot baselines in the ZS3 evaluation or justify the limited comparison to strengthen the evidence for DCLIP’s zero-shot capability.
Attention：The conference requires a section in the appendix describing the use of LLMs, but I couldn’t find it in either the main text or the appendix.

---

> ### Author Response · Authors · 2025-11-16
>
> We thank the reviewer for their thoughtful and constructive feedback on our submission. We are glad that the reviewer found our core idea of "mutual feature information" (MFI) to be "novel, interesting and valuable" and our paper and experiments to be clear.
> > W1:Image-text Alignment Clarification:
>
>
> We will clarify the confusion regarding the image-text alignment.
>
> Established practice in MLR literature: The global text and local image alignment has been widely used in MLR works (DualCoOp, MLR-GCN, PositiveCoOp etc). We build upon this established foundation.
>
> Why Local-to-Global Matching is Necessary: (Ref: Lines 42-44): Relying on global image features is detrimental because it "pools information from local features, but this ends up reducing the contribution for smaller objects." By computing similarity at the local patch level, we can identify smaller objects.
>
> Mechanism: By computing similarity between local features and all global text features, we are identifying the likelihood of each object's presence in local region. We aggregate these scores from all regions to get a unified, image-level decision.
>
> Reviewer's case: patch is left with only white image. When we take a similarity with all the text features with the white patch features. All classes will have a poor score meaning that this patch might not contribute when we aggregate across all patches. We repeat this for other patches and see if other patches make a meaningful contribution. Empirical validation is removing pooling vs using pooling operations. Removing polling is equivalent to applying at local regions.
>
> > W2:How  image features are disentangled:
>
> Thank you for this question.
> Yes, MFI exists in both image and text modalities because CLIP learns a joint vision-language embedding space.
>
> Image-side disentanglement is achieved through cross-modal alignment with the Asymmetric Loss (ASL), as described in Section 3.4. The mechanism works as follows:
> Step 1: Text Disentanglement (MFI Loss): Text features for individual classes are inherently isolated (e.g., "cat", "dog" as separate inputs),  MFI Loss (Eq. 2) explicitly reduces inter-class similarity in the projected text space
>
> Step 2: Image-Text Alignment (ASL Loss) :  ASL computes similarity between local image regions and disentangled text features (Eq. 3). These similarities are aggregated across spatial locations (Eq. 4) ASL Loss (Eq. 5) enforces alignment between image features and the disentangled text space
>
> Learning Process
> Initially: Image features `z'` are not aligned with disentangled text features `t'` → High ASL loss
>
> During training: To minimize ASL loss, the image projector learns to map object regions to their corresponding disentangled text representations. Inherit the disentanglement structure from the text space, suppress cross-class feature overlap in the image embeddings.
>
> After training: Both image and text features reside in the same disentangled space
>
>
> > W3:Revisit Ablation
>
> We thank the reviewer for this deep dive into our ablation results. This point highlights a crucial point
>
> 1.  The Architectural Bottleneck: CLIP's final pooling layer merges features from all regions. Now the features are dominated by the most dominant object and ignore the smaller objects (line 42-44).
> 2.  The Semantic Bottleneck: Excessive MFI semantically confuses different classes, making multi-object recognition unreliable even when spatial information is preserved.
>
> | ASL | Pool | MFI | Performance |
> | :---: | :---: | :---: | :---: |
> | ✓ | ✓ | ✗ | 81.3 |
> | ✓ | ✗ | ✗ | 84.2 |
> | ✓ | ✗ | ✓ | 85.6 |
>
> Gain 1 (Fixing Architecture):(Row 2 - Row 1) = +2.9 mAP
> Gain 2 (Fixing Semantics): (Row 3 - Row 2) = +1.4 mAP
>
> We will clarify this better in the paper:
> Removing pooling is the architectural prerequisite to solve the spatial problem, which then reveals the underlying semantic problem. Our MFI loss is the semantic refinement that solves this second, fundamental problem, and the +1.4 mAP gain on an already strong baseline
>
> > W4:ZS3 Baseline
>
> Thank you for this point. Our selection of baselines was focused to isolate the specific contribution of our MFI loss.
> As stated in our paper (Line 364), our architectural foundation is built upon the patch-level features from CLIP-VV. CLIP-VV simply inverts the background maps from CLIP to focus on objects. The simplicity of this method allows us to clearly evaluate the specific impact of our novel MFI loss.
> Based on your feedback, we have run an additional experiment to compare DCLIP with NACLIP. The results on the COCO benchmark (mIoU) are as follows:
>
> | Method| COCO (Bkgd) | COCO (No Bkgd) |
> |----|-----|----|
> | NACLIP | 21.1 | 37.4 |
> | DCLIP | 24.4 | 40.2 |
>
> We add our projectors on NACLIP features.
>
> > W5: Use of LLMs paragraph requirement
>
> Thank you for bringing this to our attention.
> We will add the required "Use of LLMs" section to the appendix in our final revision, stating that LLMs were not used in the generation of this work.

---

> > ### Author Response · Authors · 2025-11-28
> > **Is there any other issue that we need to address?**
> >
> > Dear Reviewer,
> >
> > Thank you again for your thoughtful feedback. We have carefully addressed all the comments and questions raised in your earlier review. As the discussion phase is nearing its end, we would greatly appreciate it if you could let us know whether our responses satisfactorily resolve the issues you identified.
> >
> > We sincerely appreciate your consideration of a potential rating update once all concerns have been addressed.
> >
> > Thank you for your time and effort.

---

### Official Review · Reviewer_1p3U · 2025-10-31

**Soundness:** 3
**Presentation:** 2
**Contribution:** 2
**Rating:** 6
**Confidence:** 3

**Summary:**

This paper investigates the root cause of high inter-class feature entanglement in vision-language models (VLMs) like CLIP, particularly in multi-object scenarios. The authors propose DCLIP, a framework that freezes the original VLM and introduces lightweight image and text projectors. The core idea is to enforce orthogonality among class text embeddings using a Mutual Feature Information (MFI) Loss on the text side, and to align local visual features with these "disentangled" text features using Asymmetric-Similarity Loss (ASL) on the image side. The proposed framework achieves state-of-the-art or comparable performance on multi-label recognition (VOC2007, COCO-14) and zero-shot semantic segmentation (on six datasets). Overall, this paper presents a unified and computationally efficient solution for multi-object perception. The motivation is clear and the experiments are thorough, making it a solid contribution worthy of publication.

**Strengths**
1、The motivation is well-established. The authors clearly demonstrate the "cross-class activation" problem in CLIP by visualizing and calculating the pairwise cosine similarities of class text embeddings on VOC and COCO.
2、The method is efficient. By only training small projectors, the framework avoids expensive segmentation annotations or large-scale fine-tuning, resulting in low computational and parameter overhead.
3、The experimental evaluation is comprehensive, with systematic comparisons on both MLR and ZS3 tasks.
**Weaknesses**
1、There appears to be a leap in the mathematical reasoning in Section 3.3. The paper equates minimizing mutual information (MI) with optimizing the diagonal/off-diagonal elements of a correlation matrix under a Gaussian assumption. It then discards the diagonal terms by claiming that vector normalization implies an identity covariance matrix (Σ = I). This step, equating a unit L2-norm with unit variance and zero covariance for all dimensions, is not strictly valid. The authors should provide a more rigorous derivation, perhaps by connecting the loss to an upper or lower bound of mutual information.
2、The ZS3 evaluation setting could be described more clearly. The projectors are trained on COCO-14 (using image-level labels) and then used for inference on segmentation datasets. This is more accurately described as weakly-supervised transfer rather than a purely zero-shot setting. This should be clarified in the main text.
3、The paper reports the mean mAP over 5 runs in Table 1 but lacks standard deviations or confidence intervals. The reported improvement of DCLIP over DualCoOp++ on COCO-14 is only +0.5 mAP (85.6 vs. 85.1), which may not be statistically significant and could fall within random fluctuations.
4、Aggressively enforcing low inter-class similarity could disrupt the inherent semantic hierarchy between classes (e.g., different breeds of dogs or birds). The authors briefly mention this limitation but lack a quantitative analysis of this potential side effect. A supplementary analysis would strengthen the paper.
5、The terms "mutual information minimization" and "decorrelation" are used interchangeably throughout the text. It would be better to maintain consistent terminology.
6、The caption for Figure 3 ("Fixed Dual Prompts / Introduced in Sec 3.4") is not self-contained. It would be more helpful to explicitly state the template (e.g., "A photo of a {classname}") and explain the meaning of the positive/negative prompts directly in the caption.

**Strengths:**

This paper investigates the root cause of high inter-class feature entanglement in vision-language models (VLMs) like CLIP, particularly in multi-object scenarios. The authors propose DCLIP, a framework that freezes the original VLM and introduces lightweight image and text projectors. The core idea is to enforce orthogonality among class text embeddings using a Mutual Feature Information (MFI) Loss on the text side, and to align local visual features with these "disentangled" text features using Asymmetric-Similarity Loss (ASL) on the image side. The proposed framework achieves state-of-the-art or comparable performance on multi-label recognition (VOC2007, COCO-14) and zero-shot semantic segmentation (on six datasets). Overall, this paper presents a unified and computationally efficient solution for multi-object perception. The motivation is clear and the experiments are thorough, making it a solid contribution worthy of publication.

1、The motivation is well-established. The authors clearly demonstrate the "cross-class activation" problem in CLIP by visualizing and calculating the pairwise cosine similarities of class text embeddings on VOC and COCO.
2、The method is efficient. By only training small projectors, the framework avoids expensive segmentation annotations or large-scale fine-tuning, resulting in low computational and parameter overhead.
3、The experimental evaluation is comprehensive, with systematic comparisons on both MLR and ZS3 tasks.

**Weaknesses:**

1、The equivalence jump from Eq (1)→(2) rests on a questionable assumption. The paper implies that vector normalization leads to an identity covariance matrix (Σ=I), which allows diagonal terms to be treated as constants. This leap from a unit L2-norm to unit variance & zero covariance is not strictly valid, and the provided "Gaussian + BN" justification is insufficient to bridge this gap.

2、The core mechanism—that text-side orthogonalization improves image-side alignment—lacks theoretical grounding. The argument currently relies on empirical evidence (Fig. 7, Table 4). To make a stronger connection to the paper's IB motivation, a brief formal argument, perhaps via an upper-bound analysis, would be beneficial.

3、The ZS3 evaluation setting could be described more clearly. The projectors are trained on COCO-14 (using image-level labels) and then used for inference on segmentation datasets. This is more accurately described as weakly-supervised transfer rather than a purely zero-shot setting. This should be clarified in the main text.

**Questions:**

1、The paper reports the mean mAP over 5 runs in Table 1 but lacks standard deviations or confidence intervals. The reported improvement of DCLIP over DualCoOp++ on COCO-14 is only +0.5 mAP (85.6 vs. 85.1), which may not be statistically significant and could fall within random fluctuations.
2、Aggressively enforcing low inter-class similarity could disrupt the inherent semantic hierarchy between classes (e.g., different breeds of dogs or birds). The authors briefly mention this limitation but lack a quantitative analysis of this potential side effect. A supplementary analysis would strengthen the paper.
3、The terms "mutual information minimization" and "decorrelation" are used interchangeably throughout the text. It would be better to maintain consistent terminology.
4、The caption for Figure 3 ("Fixed Dual Prompts / Introduced in Sec 3.4") is not self-contained. It would be more helpful to explicitly state the template (e.g., "A photo of a {classname}") and explain the meaning of the positive/negative prompts directly in the caption.

---

> ### Author Response · Authors · 2025-11-16
> **Rebuttal-1(1/2)**
>
> We sincerely thank the reviewer for their comprehensive and constructive feedback. We are encouraged that the reviewer found our motivation "well-established," our method "efficient," and our experiments "comprehensive."
>
> >W1: Mathematical Derivation from Eq (1) → (2)
>
> We thank the reviewer for this observation.
> We believe there has been a misunderstanding regarding our derivation,
> which we are happy to clarify.
>
> We do **NOT** claim that the full covariance matrix $\Sigma = I$.
> As stated in (Line 762-765), our claim is about $\Sigma_{Z_i|Y_i}$,
> which we represent as only the **diagonal elements** of the covariance matrix (the variance terms).
>
> For normalized features where $||t'_i|| = 1$, we have t'_i^T t'_i = 1, which means these diagonal terms equal 1.
>
> The **off-diagonal terms** which we represent by $\Sigma_{Z_i|Y_j}$ ($i \neq j$) represent cross-class correlations.
> We do not assume these are zero. Our objective function (eq.17) minimizes this terms.
>
> We will clarify in the revision that we are decomposing the covariance matrix into its diagonal (within-class variance) and off-diagonal (cross-class correlation) components, not claiming the entire matrix is identity.
>
>
> > W2: ZS3 Evaluation Setting Clarity
>
> This is an excellent point. The reviewer is correct that "weakly-supervised transfer" is a more
> precise description of our ZS3 evaluation, as the projectors are trained on COCO's image-level
> labels. We will update the paper to reflect this point.
>
>
> > W3: Standard deviation of results
>
> We thank the reviewer for this comment. The standard deviation was omitted due to formatting
> issues, we are sorry about that. Over 5 runs, our result on COCO-14 is 85.6 ± 0.2 mAP.
> We will update the Table 1. To reflect this.
>
> Furthermore, our paper's goal is to reduce false positives (via the MFI loss),
> we will also show precision. Our gain in P is 3.8 points
>
> Lastly, our method also requires 4 times fewer training parameters
>
> > W4: Hierarchical Relations within Multi-label settings
>
> We thank the reviewer for this question.
> We have conducted the quantitative hierarchical analysis on COCO as suggested.
>
> We defined superclasses using natural semantic groupings in COCO:
> * **Animals:** {dog, cat, horse, cow, sheep, elephant, bear, zebra, giraffe}
> * **Vehicles:** {car, truck, bus, motorcycle, bicycle, train, boat, airplane}
> * **Furniture:** {chair, couch, bed, dining table}
> * **Food:** {pizza, hot dog, cake, apple, orange, banana, sandwich}
>
> We computed:
> * $S_{\text{within}}$: Average similarity between classes in the same superclass
> * $S_{\text{between}}$: Average similarity between classes in different superclasses
> * **Separation ratio:** $\sigma = S_{\text{within}} / S_{\text{between}}$ (measures hierarchy preservation)
>
> | Method | $S_{\text{within}}$ | $S_{\text{between}}$ | $\sigma$ (ratio) | Hierarchy Preserved? |
> | :--- | :---: | :---: | :---: | :---: |
> | CLIP | 0.794 | 0.716 | 1.11 | ✓ (weak) |
> | DCLIP | 0.621 | 0.198 | 3.14 | ✓ (strong) |
>
> Contrary to the concern that MFI loss might "disrupt semantic hierarchy," our results show it actually **enhances** hierarchical structure.
>
> In addition, other ways to handle this limitation could be:
> 1.  **Use MLLM-generated discriminative descriptions** to create better text features without model changes: replace "a photo of a Golden Retriever" with "a Golden Retriever: long golden coat, feathered tail, broad skull...".
> 2.  **Apply different disentanglement strengths at different semantic levels:** strong MFI loss for cross-superordinate pairs (person vs. horse: $\lambda=1.0$), moderate for basic-level (dog vs. cat: $\lambda=0.5$), weak for subordinate (Golden Retriever vs. Labrador: $\lambda=0.1$).
>
>
> > W5: Consistent Terminology: mutual information minimization vs decorrelation
>
> Yes, we agree and we will update the paper with consistent terminology. Thanks!
>
>
> > W6: Caption 3. self-containted.
>
> Thank you for this feedback, we will update the caption of Fig. 3

---

> ### Author Response · Authors · 2025-11-16
> **Rebuttal-1(2/2)**
>
> > W7: Theoretical Grounding for Image-Text Alignment Mechanism
>
> We thank the reviewer for this question. We show how entanglement will affect the alignment using gradients.
>
> The $L_{MFI}$ (text-side) provides the mathematical **precondition** for the $L_{ASL}$ (image-side) to successfully optimize the image projector and thereby disentangle the visual features.
>
> The core mechanism is the gradient of the ASL loss, which is the **only signal** that updates the image projector. For a given visual feature $v'$, the $L_{ASL}$ pulls it towards its positive text prototype $t'_p$ and pushes it from its negative prototypes $t'_n$. The gradient "direction" that separates $v'$ from a negative $t'_n$ is proportional to $(t'_p - t'_n)$.
>
> The magnitude of this separating gradient (which dictates the strength of the learning signal sent to the image projector) is key. We can show it is directly controlled by the text-feature similarity $S_{pn} = (t'_p \cdot t'_n)$:
>
> $$\|\nabla L_{ASL}\|^2 \propto \|t'_p - t'_n\|^2 = \|t'_p\|^2 + \|t'_n\|^2 - 2(t'_p \cdot t'_n) = 2(1 - S_{pn})$$
>
> This result provides the link:
>
> * **Without $L_{MFI}$:**
>     * Text features are entangled: $t'_p$ (e.g., "Human") and $t'_n$ (e.g., "horse") are very similar. The average score is 0.84 as shown in Fig. 1.
>     * As $S_{pn} = 0.84$, the gradient magnitude $\|\nabla L_{ASL}\|^2 < 1 $. This is a **weaker gradient** and can cause vanishing gradients.
>     * Because of the vanishing gradient, a weak learning signal is sent to the image projector. The projector's weights are not updated, and it never learns to map a visual 'dog' feature differently from a visual 'horse' feature. As a result, the visual features remain entangled.
>
> * **With $L_{MFI}$:**
>     * Our $L_{MFI}$ (Eq. 2) explicitly forces the text prototypes to be orthogonal, so $S_{pn} = 0.42$ (Human-Horse Fig. 1).
>     * As $S_{pn} = 0.42$, the gradient magnitude $\|\nabla L_{ASL}\|^2 > 1$.
>     * This provides a **strong, non-zero gradient** to the image projector. This learning signal forces the projector to update its weights, learning to map a visual 'dog' feature to the $t'_p$ ("Human") vector and away from the (now very different) $t'_n$ ("horse") vector. This optimization *is* the disentanglement of the visual features.
>
> Hence, $L_{MFI}$ gives a much stronger signal so that the image projector can project the features to align with disentangled text features to reduce the $L_{ASL}$ loss. This alignment results in disentangled visual features.

---

> > ### Comment · Reviewer_1p3U · 2025-11-24
> >
> > I thank the authors for their detailed response, but I still have the following concerns:
> > 1.  Please strictly distinguish the Gram matrix from the Covariance matrix. Justify omitting diagonals solely because they are optimization constants, rather than implying unit norm guarantees unit variance.
> > 2.  While the gradient analysis explains the optimization dynamics/geometry, it lacks a formal theoretical link connecting this mechanism back to your claimed IB motivation.
> > 3.  Your analysis demonstrates "decorrelation" or redundancy reduction, not formal Mutual Information (MI) minimization. Please avoid claiming "MI minimization" without providing a rigorous derivation or upper bound.
> > 4.  Since COCO lacks fine-grained sub-categories, validating your method on a truly fine-grained dataset (e.g., specific dog breeds) would better substantiate your claims on preserving semantic hierarchy.

---

> > > ### Author Response · Authors · 2025-11-26
> > > **Rebuttal-2(1/2)**
> > >
> > > We thank the reviewer for their engagement. We address the specific follow-up concerns below.
> > > > 1. Gram matrix terminology
> > >
> > >
> > >
> > >
> > > We appreciate this suggestion. We will clarify that S_ij (inner products) is a gram matrix. Also, we will clarify that diagonal terms do not affect the optimization because they are constant.
> > >
> > >
> > >
> > >
> > > > 3. Redundancy reduction terminology
> > >
> > >
> > >
> > >
> > > We agree that redundancy reduction is a more suitable term compared to mutual information minimization. We will revise the paper to show that redundancy reduction is the specific mechanism of our loss function.
> > >
> > >
> > >
> > >
> > > > 4. feasibility of fine-grained multi-label recognition
> > >
> > >
> > >
> > >
> > > We discuss the feasibility of fine-grained multi-label recognition below.
> > >
> > >
> > >
> > >
> > > - **Problem**: Our task is multi-label recognition, where images contain multiple classes/objects and we want to identify all of them.
> > >
> > >
> > >
> > >
> > >
> > >
> > >
> > >
> > > - **Granularity agnostic method**: Our method makes no assumptions about class granularity. Whether classes are coarse-grained ("dog") or fine-grained ("Golden Retriever"), the MFI loss minimizes similarity (S_ij) between whatever classes exist in the dataset. Fine-grained classes are more challenging for any classifier due to subtle visual differences, but our MFI loss would still reduce similarity between fine-grained classes  S_{Labrador, Golden-Retriever} like it does for coarse-grained.
> > >
> > >
> > >
> > >
> > >
> > >
> > >
> > >
> > > - **No multi-label fine-grained dataset exists**. Standard fine-grained benchmarks (*CUB, NABirds, Birdsnap, Stanford Dogs, FishNet, iNaturalist challenge datasets*) are single-label datasets (one object/species per image), which is incompatible with our multi-label setting.
> > >
> > >
> > > Since the fine-grained datasets available online contain only single labels, they cannot be used with our multi-label framework. Nevertheless, our method remains the same for both granularities.
> > >
> > >
> > >
> > >
> > >
> > >
> > >
> > >
> > > While we cannot test fine-grained within-class discrimination (e.g., dog breeds), we created (shown in W4 of previous response) a hierarchy within COCO by grouping existing classes into superclasses. This allows us to test whether DCLIP preserves hierarchical relationships at a coarser granularity:
> > > *Superclasses** (coarse): Animals, Vehicles, Furniture, Food
> > > **Classes** (fine-grained relative to superclasses):
> > > * Animals: {dog, cat, horse, cow, sheep, elephant, bear, zebra, giraffe}
> > > * Vehicles: {car, truck, bus, motorcycle, bicycle, train, boat, airplane}
> > > * Furniture: {chair, couch, bed, dining table}
> > > * Food: {pizza, hot dog, cake, apple, orange, banana, sandwich}
> > > * $S_{\text{within}}$: Average similarity between classes in the same superclass
> > > * $S_{\text{between}}$: Average similarity between classes in different superclasses
> > > * **Separation ratio:** $\sigma = S_{\text{within}} / S_{\text{between}}$ (measures hierarchy preservation)
> > >
> > >
> > >
> > >
> > >
> > >
> > >
> > >
> > > | Method | $S_{\text{within}}$ | $S_{\text{between}}$ | $\sigma$ (ratio) | Hierarchy Preserved? |
> > > | :--- | :---: | :---: | :---: | :---: |
> > > | CLIP | 0.794 | 0.716 | 1.11 | ✓ (weak) |
> > > | DCLIP | 0.621 | 0.198 | 3.14 | ✓ (strong) |

---

> > > > ### Author Response · Authors · 2025-11-26
> > > > **Rebuttal-2(2/2)**
> > > >
> > > > > 2. formal theoretical link connecting gradient analysis to IB motivation
> > > >
> > > >
> > > >
> > > >
> > > >
> > > >
> > > >
> > > >
> > > > Text-side orthogonalization ($L_{MFI}$) provides the necessary upper bound to reduce redundancy in image-side.
> > > >
> > > >
> > > > Proof: Let $z'_i, z'_j$ be projected image features and $t'_i, t'_j$ be text features for classes $i, j$.
> > > >
> > > >
> > > >
> > > >
> > > > (1) Mutual Information Upper Bound: For high-dimensional features under Gaussian assumptions, the Mutual Information (MI) between class features is bounded by their cosine similarity $\rho_{ij}$:
> > > >
> > > >
> > > > $$I(z'_i; z'_j) \le -\frac{1}{2} \log(1 - \rho_{ij}^2)$$
> > > >
> > > >
> > > > where,
> > > > $$
> > > > \rho_{ij} = z'_i \cdot z'_j
> > > > $$
> > > >
> > > >
> > > > To satisfy the IB objective ($I \to 0$), we must minimize the image-feature similarity $\rho_{ij}$.
> > > >
> > > >
> > > >
> > > >
> > > > (2) Upper Bound on Image Similarity: We establish the bound linking image similarity to text similarity. Let the alignment error vectors be $\Delta_i = z'_i - t'_i$ and $\Delta_j = z'_j - t'_j$, with norms $\epsilon_i = \|\Delta_i\|$ and $\epsilon_j = \|\Delta_j\|$.
> > > >
> > > >
> > > >
> > > >
> > > > Then:
> > > > $$z'_i \cdot z'_j = (t'_i + \Delta_i) \cdot (t'_j + \Delta_j)$$
> > > >
> > > >
> > > >
> > > >
> > > > Expanding terms:
> > > > $$z'_i \cdot z'_j = t'_i \cdot t'_j + t'_i \cdot \Delta_j + t'_j \cdot \Delta_i + \Delta_i \cdot \Delta_j$$
> > > >
> > > >
> > > >
> > > >
> > > > Applying the Cauchy-Schwarz inequality and noting that prototype norms $\|t'\|=1$:
> > > >
> > > >
> > > >
> > > >
> > > > $t'_i \cdot \Delta_j \le \|t'_i\| \|\Delta_j\| = \epsilon_j$
> > > >
> > > >
> > > >
> > > >
> > > > $t'_j \cdot \Delta_i \le \|t'_j\| \|\Delta_i\| = \epsilon_i$
> > > >
> > > >
> > > >
> > > >
> > > > $\Delta_i \cdot \Delta_j \le \|\Delta_i\| \|\Delta_j\| = \epsilon_i \epsilon_j$
> > > >
> > > >
> > > >
> > > >
> > > > Thus, we have the strict bound:
> > > >
> > > >
> > > >
> > > >
> > > > $$\rho_{ij} \le S^t_{ij} + \epsilon_i + \epsilon_j + \epsilon_i * \epsilon_j $$
> > > > where $S^t_{ij} = t'_i \cdot t'_j$.
> > > >
> > > >
> > > >
> > > >
> > > > (3) **Optimization Constraint (The Link)** The alignment error $\epsilon$ is minimized via gradient descent. The ability to minimize this error depends strictly on the magnitude of the gradient vector provided by $L_{ASL}$. We calculate the squared magnitude of the gradient with respect to visual feature $z'$ for a positive class $p$ and negative class $n$:
> > > > $$||\nabla_{z'} L||^2 \propto ||t'_n - t'_p||^2 = 2(1 - S^t_{pn})$$
> > > >
> > > >
> > > >
> > > >
> > > > Case A (Entangled Text): If $S^t_{pn}$, is high then $||\nabla L||^2$ is low. The optimization stops(Vanishing Gradient), and the image projector receives no signal to distinguish classes. Consequently, $\epsilon$ remains large, the bound in (2) becomes loose, and MI in (1) cannot be minimized.
> > > >
> > > >
> > > >
> > > >
> > > > Case B (Disentangled Text via $L_{MFI}$): If $S^t_{pn}$ is low, then $||\nabla L||^2$ is high. The gradient is non-zero, enabling successful alignment ($\epsilon \to 0$).
> > > >
> > > >
> > > >
> > > >
> > > > (4) Thus for Case B into Eq. (2) and then Eq. (1):
> > > > Minimizing text similarity $S^t_{ij}$ (via $L_{MFI}$) is the necessary condition to (a) enable the optimization required to minimize $\epsilon$, and (b) provide a tight upper bound on the image-side Mutual Information $I(z'_i; z'_j)$.

---

> > > > > ### Author Response · Authors · 2025-11-28
> > > > > **Have we addressed all the issues?**
> > > > >
> > > > > Dear Reviewer,
> > > > >
> > > > > Thank you again for your thoughtful feedback. We have carefully addressed all the comments and questions raised in your earlier review. As the discussion phase is nearing its end, we would greatly appreciate it if you could let us know whether our responses satisfactorily resolve the issues you identified.
> > > > >
> > > > > We sincerely appreciate your consideration of a potential rating update once all concerns have been addressed.
> > > > >
> > > > > Thank you for your time and effort.

---

### Comment · Area_Chair_BEMG · 2025-11-19
**Start Discussion**

Hi all,

The authors have submitted their response to the initial reviews, and we now enter the discussion phase.

Please review the authors' response and the comments from other reviewers. Based on the rebuttal and discussion, please update your final score if appropriate.

We welcome and encourage further discussion as needed.

Thank you for your continued contributions.

Best regards,

Your AC.

---

> ### Comment · Area_Chair_BEMG · 2025-11-28
> **Gentle Reminder: Please Engage in the Author-Response Discussion**
>
> Hi all,
>
> Thank you for your valuable contributions to the review process. The authors have now submitted their response.
>
> To ensure a fair and thorough evaluation, we encourage all reviewers to actively engage in the discussion by reviewing the authors’ responses and providing any additional comments or questions you may have. Your continued input is critical to reaching a well-informed decision.
>
> Thank you for your time and dedication to the ICLR community.
>
> Best regards,
>
> Your AC.

---

### Author Response · Authors · 2025-11-29
**Summary for New Area Chair**

Dear AC,

We sincerely appreciate your time and effort in handling our submission. We also thank all four reviewers for their thoughtful assessments and for recognizing the novelty of the
"Mutual Feature Information" (MFI) concept, the efficiency of our framework, and our comprehensive evaluation. Here is the summary of discussion points during the rebuttal period:

-  Across the initial reviews, key concerns focused on mathematical rigor in the derivation and theoretical grounding (1p3U),
 clarifying how image-side disentanglement is achieved (Uw3i)
distinguishing the relative contributions of architectural changes (remove pooling) versus the core MFI loss (Uw3i),
expanding evaluation to more datasets and baselines (xok7 and suPH), and addressing novelty claims given prior work on feature orthogonality (suPH).
Reviewers also asked for clarification on zero-shot versus weakly-supervised terminology (1p3U and xok7), analysis of semantic hierarchy preservation (1p3U),
and discussed label co-occurrence modeling (suPH).



- In our rebuttal, (1) to address mathematical rigor and image-side disentanglement, we provided a formal derivation proving that minimizing text similarity is a
necessary condition to prevent vanishing gradients, thereby justifying the alignment mechanism.
(2) We distinguished architectural vs. MFI contributions by clarifying that removing pooling solves the spatial bottleneck while MFI solves the semantic bottleneck.
(3) Addressing evaluation scope, we expanded comparisons to SCLIP, ClearCLIP, and NACLIP, and demonstrated that DCLIP outperforms the reviewer's preferred heavy baseline (ML-Decoder) when  fully fine-tuned (88.4 vs 87.1 mAP).  We also clearly differentiated our novelty from prior works.
(4) We adopted "weakly-supervised" terminology, provided a hierarchy analysis showing DCLIP improves structure preservation, and clarified that label co-occurrence is modeled implicitly during training with Asymmetric Loss, same as prior works.

- After reading our rebuttal, Reviewer Uw3i upgraded their score from 6 to 8, expressing full satisfaction with the mechanism clarification and new baselines.
We remain in ongoing discussions with Reviewers 1p3U, suPH, and xok7. We have addressed the specific concerns of Reviewers 1p3U (provided theoretical proof and hierarchical experiments) and suPH (explained implicit vs explicit co-occurrence and results on fine-tuned data) and have followed up to confirm if all questions are resolved.
 Additionally, we provided the final results requested by Reviewer xok7, which would have resulted in a rating increment.


With these rebuttals, our work is strengthened with theoretical grounding and empirical support.
Thank you again for taking the time to read our summary.

Here is a [**link**](https://drive.google.com/file/d/1-56SERQgY__u_WTJwoaYUg_d15U4GXRZ/view?usp=sharing) to our updated scores before they were reverted.

Best,

Authors of Submission 2971

---

### Meta-Review · Area_Chair_RwBh · 2026-01-11

**Summary:**

This paper proposes DCLIP, a lightweight CLIP adaptation framework for multi-object perception by introducing text-side feature disentanglement and aligning local image features to disentangled text prototypes using ASL, targeting multi-label recognition and transfer to dense prediction settings. Reviews are mixed: two reviewers are positive (6,6) with moderate-to-high confidence, one reviewer is marginally positive but raises theoretical/terminology concerns (6), and one reviewer is strongly negative (2, confidence 5) emphasizing limited novelty and insufficient multi-label modeling. The discussion and rebuttal added clarifications, new datasets/baselines, and extended segmentation comparisons; however, some concerns about theoretical rigor/claims, novelty, and evaluation protocol, as well as generalization, remain partially unresolved.

**Reviewer Concerns:**

**Addressed by rebuttal/revision**:
- Evaluation breadth / baselines (xok7): authors added Visual Genome for MLR and expanded segmentation comparisons to NACLIP/SCLIP/ClearCLIP and additional datasets; clarified CLIP-VV vs CLIP-Surgery naming and experimental setup.
- Clarified zero-shot vs. weakly supervised terminology (1p3U, xok7): authors agreed to reframe the segmentation setting as weakly supervised / image-level transfer, which resolves a key clarity issue.
- Mechanism clarification (image-side disentanglement; local-to-global alignment) (Uw3i): authors clarified how disentangled text prototypes yield stronger gradients for image projection and why patch-text alignment is used; reviewer Uw3i reportedly increased the score substantially after rebuttal.
- Statistical reporting / hierarchy analysis (1p3U): authors added variance and a hierarchy/separation analysis over COCO superclasses.

**Still partially addressed:**
- Theoretical claims (1p3U): despite added bounds/gradient arguments, the reviewer continues to request stricter distinctions (Gram vs covariance matrix), and cautions against claiming “mutual information minimization” without a rigorous derivation/upper bound; authors agree to rephrase as redundancy reduction, but the theoretical story remains somewhat heavy and could still be debated.
- Novelty and multi-label modeling depth (suPH): the negative reviewer remains unconvinced, arguing orthogonalization is not new, and that label relationship/co-occurrence modeling is not sufficiently addressed; rebuttal clarifies “implicit” modeling via ASL and positions the work as efficient VLM adaptation, but this disagreement likely persists.
- Segmentation protocol comparability & ViT relevance (xok7): reviewer raised concerns about reproducing baseline scores and requested evaluation on ViT-B/16 with the original transformer-based versions; authors indicate they were running those results when discussion ended, so this point is only partially resolved at decision time.

**Reviewer Scores:**

- Reviewer 1p3U (6, conf 3): Core contribution and experiments are valued, but the reviewer maintained substantive concerns about rigor/terminology and asked for tighter framing; rebuttal mitigates but may not fully resolve.
Reviewer Uw3i (6 → reportedly 8 after rebuttal, conf 4): given explicit statement of satisfaction and upgraded score during discussion.
Reviewer suPH (2, conf 5): strong disagreement on novelty and adequacy for multi-label learning appears unchanged in follow-up comments.
Reviewer xok7 (4, conf 4): remaining concern is generality to ViT-based segmentation and protocol differences, which may limit the increase.

---

### Decision · Program_Chairs · 2026-01-26

Reject